# Excitonic topology and quantum geometry in organic semiconductors

Wojciech J. Jankowski [1] ✉, Joshua J. P. Thompson[2], Bartomeu Monserrat [1,2] & Robert-Jan Slager [1,3] ✉

Excitons drive the optoelectronic properties of organic semiconductors which underpin devices including solar cells and light-emitting diodes. Here we show that excitons can exhibit topologically non-trivial states protected by inversion symmetry and identify a family of organic semiconductors realising the predicted excitonic topological phases. We also demonstrate that the topological phase can be controlled through experimentally realisable strains and chemical functionalisation of the material. Appealing to quantum Riemannian geometry, we predict that topologically non-trivial excitons have a lower bound on their centre-of-mass spatial spread, which can significantly exceed the size of a unit cell. Furthermore, we show that the dielectric environment allows control over the excitonic quantum geometry. The discovery of excitonic topology and excitonic Riemannian geometry in organic materials brings together two mature fields and suggests many new possibilities for a range of future optoelectronic applications.

Topology represents a versatile tool that drives the study of diverse condensed matter phenomena. Topological invariants arise due to phases acquired by wave functions when adiabatically transported around the Brillouin zone (BZ), and provide a classification of states of matter that has radically transformed our understanding of inorganic materials over the past few decades[1,2]. This work has culminated in a rather complete understanding of free fermionic band structures, which can be characterised through the gluing of symmetry eigenvalues, or irreducible representations of eigenstates, between high symmetry points in the momentum space Brillouin zone[3–7]. Assessing whether this admits an exponentially localised Wannier representation in real space leads to their topological characterisation[8,9]. Current efforts are moving from free fermion systems towards the question of how these ideas can drive new understanding in interacting systems. An example of an interacting system is that of electron-hole bound states, or excitons.

Excitons dominate the optoelectronic properties of a wide range of materials, particularly low-dimensional materials and materials composed of organic molecules. Organic semiconductors, in particular, possess excellent light-harvesting properties driven by the formation of excitons[10–13], while being chemically versatile[12,14], cheap, and environmentally friendly to fabricate[15,16]. These features make organic semiconductors one of the most promising material platforms in which to realise optoelectronic devices, from photovoltaics[17,18] and light-emitting diodes[19,20] to biosensors[21]. The electron-hole distance and the centre-of-mass location of excitons can differ greatly between materials, and delocalised excitons with high mobilities are particularly promising for applications[22].

In this work, we establish a link between the rich exciton phenomenology in organic semiconductors with topological insights of such phases, developing a promising platform in which these ideas could culminate in experimentally viable topological phases and phenomena. In particular, we fully characterise the inversion-symmetry-protected excitonic topology in one-dimensional systems in terms of the excitonic Berry phase and a $\mathbb{Z}_2$ excitonic topological invariant. Additionally, we derive a lower bound on the spread of the exciton centre-of-mass wavefunction arising from a non-trivial excitonic quantum metric induced by the excitonic topology. We derive these results using model Hamiltonians and, importantly, we also predict their manifestation in the polyacene

[1]TCM Group, Cavendish Laboratory, Department of Physics, Cambridge, UK. [2]Department of Materials Science and Metallurgy, University of Cambridge, Cambridge, UK. [3]Department of Physics and Astronomy, University of Manchester, Oxford Road, Manchester, UK. ✉e-mail: wjj25@cam.ac.uk; rjs269@cam.ac.uk

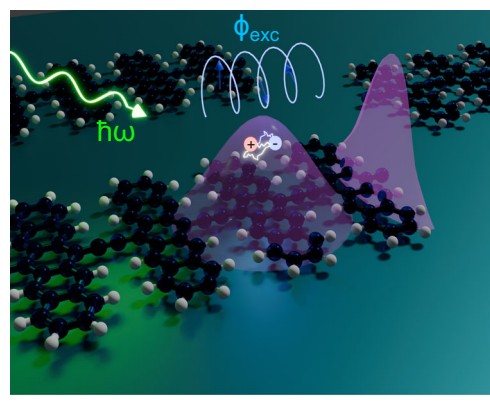

**Fig. 1 | Topological excitons in organic polymers.** Formation of topological excitons (purple) from electron (blue) and hole (red) pairs in organic materials, induced by photoexcitation with photons of energy $\hbar\omega$ (green). Topological excitons exhibit a winding of the excitonic Bloch vectors $|u_{\mathbf{Q}}^{\text{exc}}\rangle$ as the momentum of the excitons is changed, which is captured by the excitonic Berry phases $\phi_{\text{exc}}$ and associated inversion symmetry-protected topological $\mathbb{Z}_2$ invariant $P_{\text{exc}}$.

family of one-dimensional organic semiconductors. We identify material realisations of predicted topological excitonic phases, which can be manipulated both chemically through the lateral size of the polyacene chains and externally through strain and through changes in the dielectric environment. Our findings of exotic excitonic states in organic materials bring together organic chemistry, semiconductor physics, and topological condensed matter, as illustrated in Fig. 1.

## Results

### Excitonic topology and Berry phases

The topological character of excitons is fully encoded in the excitonic wavefunction $|\psi_{\mathbf{Q}}^{\text{exc}}\rangle$. In a periodic crystal, the excitonic wavefunction[23–27] can be decomposed in terms of (i) electronic single-particle Bloch states $|\psi_{\mathbf{k}}^{\text{e}}\rangle = e^{i\mathbf{k}\cdot\mathbf{r}_e}|u_{\mathbf{k}}^{\text{e}}\rangle$, (ii) hole single-particle Bloch states $|\psi_{\mathbf{k}}^{\text{h}}\rangle = e^{i\mathbf{k}\cdot\mathbf{r}_h}|u_{\mathbf{k}}^{\text{h}}\rangle$, and (iii) an interaction-dependent envelope function $\psi_{\mathbf{Q}}(\mathbf{k})$; leading to a coherent superposition of electron-hole pairs:

$$\left|\psi_{\mathbf{Q}}^{\text{exc}}\right\rangle = \sum_{\mathbf{k}} \psi_{\mathbf{Q}}(\mathbf{k}) e^{i\mathbf{k}\cdot(\mathbf{r}_e - \mathbf{r}_h)} \left|u_{\mathbf{k}+\mathbf{Q}/2}^{\text{e}}\right\rangle\left|u_{-\mathbf{k}+\mathbf{Q}/2}^{\text{h}}\right\rangle. \tag{1}$$

We also introduce $|u_{\mathbf{Q}}^{\text{exc}}\rangle$ as the cell-periodic part of the excitonic Bloch state $|\psi_{\mathbf{Q}}^{\text{exc}}\rangle = e^{i\mathbf{Q}\cdot\mathbf{R}}|u_{\mathbf{Q}}^{\text{exc}}\rangle$, where $\mathbf{R} = (\mathbf{r}_e + \mathbf{r}_h)/2$ is the centre-of-mass position of the exciton. Notably, the relative position $\mathbf{r} = \mathbf{r}_e - \mathbf{r}_h$ of the electron and hole enters the excitonic state via phase factors weighted by the envelope amplitudes $\psi_{\mathbf{Q}}(\mathbf{k})$. The solutions of the single-particle electronic problem for electrons and holes, and of a two-body equation for the envelope part, fully determine the excitonic wavefunction. The excitonic topology can then be studied from the cell-periodic part $|u_{\mathbf{Q}}^{\text{exc}}\rangle$, on eliminating the phase factor with the centre-of-mass momentum $\mathbf{Q}$ coupling to the centre-of-mass position $\mathbf{R}$.

Focusing on one spatial dimension and on the associated one-dimensional momentum space with exciton momenta $\mathbf{Q} = Q$, K-theory[5,28], which provides classifications that are stable up to adding an arbitrary number of trivial bands to the system, dictates that in the presence of inversion symmetry we can obtain an excitonic $\mathbb{Z}_2$ invariant $P_{\text{exc}} = \phi_{\text{exc}}/\pi \bmod 2$ from a Berry phase[29]:

$$\phi_{\text{exc}} = \int_{\text{BZ}} \mathrm{d}Q\, A_{\text{exc}}(Q), \tag{2}$$

where $A_{\text{exc}}(Q) = i\langle u_Q^{\text{exc}}|\partial_Q u_Q^{\text{exc}}\rangle$ is the excitonic Berry connection. Alternatively, the excitonic $\mathbb{Z}_2$ invariant can be rewritten as[30]:

$$P_{\text{exc}} = \frac{1}{\pi} \int_{\text{hBZ}} \mathrm{d}Q\, [A_{\text{exc}}(Q) + A_{\text{exc}}(-Q)], \tag{3}$$

with the integration performed over a half Brillouin zone (hBZ) because the inversion symmetry relates the excitonic bands at $Q$ and $-Q$. The $\mathbb{Z}_2$ nature of the invariant can be understood intuitively: at the inversion-symmetry-invariant momenta, $Q = 0$ and $Q = \pi$, the excitonic eigenvectors $|u_Q^{\text{exc}}\rangle$ have the same relative phase in the trivial excitonic regime, whereas in the topological regime these have opposite inversion eigenvalues related by a $\pi$ phase. The invariant $P_{\text{exc}}$ precisely distinguishes two different topologies which are not transversable from one to another without closing a gap in the excitonic bands, similarly to the inversion-protected topology of electrons necessitating a gap closure to change the Berry phase, and for a topological phase transition to occur.

### Modelling excitons in one dimension

To explore excitonic topology in a one-dimensional setting, we consider the excitonic Wannier equation for the envelope function $\psi_Q(k)$:

$$\sum_{k'} h_{k,k'}(Q)\psi_Q(k) = E(Q)\psi_Q(k), \tag{4}$$

where $E(Q)$ is the exciton binding energy and $h_{k,k'}(Q) = (E_{k+Q/2}^e - E_{k-Q/2}^h)\delta_{k,k'} - W_{k,-k',Q}$ includes the single-particle energies of the electron $E_{k+Q/2}^e$ and hole $E_{k-Q/2}^h$ bands, and the electron-hole interaction $W_{k,-k',Q}$.

For the single-particle electron and hole states, which we emphasise, only form a part of the model, we use the Su-Schrieffer-Heeger (SSH) model[31,32]:

$$H = -t_1 \sum_j c_{B,j}^\dagger c_{A,j} - t_2 \sum_j c_{B+1,j}^\dagger c_{A,j} + \text{h.c.}, \tag{5}$$

with $c_{B,j}^\dagger/c_{A,j}$ the creation/annihilation operators for the electrons at sublattices $A$, $B$, in unit cell $j$, and effective staggered hopping parameters $t_1$ and $t_2$. We define the origin of a unit cell such that $t_1$ is the intracell hopping and $t_2$ is the intercell hopping across the boundary of the unit cell.

### Characterisation of topological excitons

By numerically solving the Wannier equation, we construct the excitonic phase diagram of our fully interacting problem as a function of single-particle states characterised by the hopping parameters $t_1$ and $t_2$, as shown in Fig. 2a. The phase diagram exhibits two different regimes, one corresponding to trivial excitonic topology and the other corresponding to non-trivial excitonic topology.

In regime I, corresponding to $t_1 > t_2$ in Fig. 2, we have trivial electrons and holes yielding trivial excitons. This is a consequence of the absence of winding in the vectors $|u_k^e\rangle$ and $|u_k^h\rangle$ that leads to a trivial electronic Berry phase with $\phi_e = 0$. This regime is associated with both electron and excitonic Wannier centres localised at the centres of the unit cells, see Fig. 3a.

In regime II, corresponding to $t_1 < t_2$, we have topological electrons and holes, resulting in topological excitons, whose topological character is inherited from the underlying electron and hole topology. Explicitly, we have that the single-particle Berry phases $\phi_e$ for electrons, $\phi_h$ for holes, and the excitonic Berry phase $\phi_{\text{exc}}$, obey $\phi_{\text{exc}} = \phi_e = \phi_h = \pi$. This regime is associated with both electron and exciton Wannier centres localised at the edges of the unit cells, see Fig. 3b.

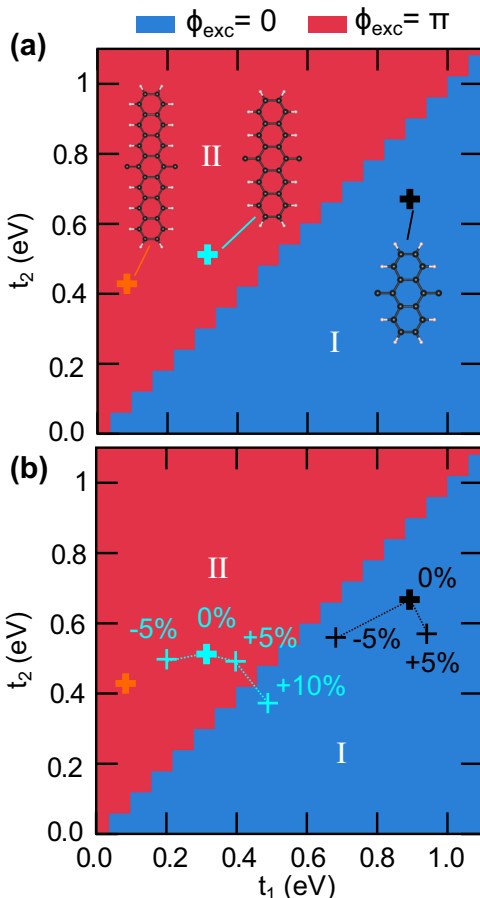

**Fig. 2 | Topological excitonic phase diagrams. a** Phase diagram showing excitonic topological phases as a function of $t_1$ and $t_2$, with $L = 0.8$ nm and $\alpha_{1D} = 0.05$ eVÅ, see Methods. The phase regimes are defined as I, II, as discussed in the main text. We also indicate the location of polyanthracene (black cross), polypentacene (cyan cross) and polyheptacene (orange cross) in the phase diagram. **b** Controllability of excitonic topology with strain. We explore the excitonic topological phase diagram of the lowest excitonic band on applying uniform strains within a range $\gamma = \pm10\%$ of the relative polymer length change. We find that the excitonic phases can evolve from region II to region I, demonstrating that the non-trivial excitonic topology can be trivialised with strain $\gamma < 10\%$.

## Phenomenological understanding of the phase diagram

To gain a deeper understanding of the excitonic topology and phase diagram presented in Figs. 2 and 3, we now show that the electron-hole interaction can be phenomenologically captured through a dualisation picture using a SSH-Hubbard-like model with a hierarchy of electronic density-density interactions $U_i$, with $i = 1, 2, 3, \ldots$ labelling interactions between $n$th order left neighbours for odd $i$, and $n$th order right neighbours for even $i$ (see Methods). The weak and strong interaction limits of this Hamiltonian provide a simple picture that allows us to rationalise the results presented in Figs. 2 and 3.

In the first limit we consider an interaction smaller than the bandwidth $U_i \ll t_1, t_2$, obtaining the standard SSH model[31,32]. For $t_2 < t_1$, we obtain vanishing Berry phases for electrons, holes, and excitons, with all of them localised at the origin of the unit cell [Fig. 3a], and corresponding to region I of Fig. 2. For $t_2 > t_1$, and in this dispersive limit, the electrons, holes, and as a consequence, the excitons, all acquire $\pi$ Berry phases. This is also reflected by the inversion-symmetry protected shift of all with respect to the unit cell origin [Fig. 3b], corresponding to region II of Fig. 2.

The other limit we consider corresponds to dominant interactions. This limit is characterised by flat bands ($t_2 \gg t_1$ or $t_2 \ll t_1$) and the interactions take the dominant role $U_i \gg t_1, t_2$. We focus on semilocal

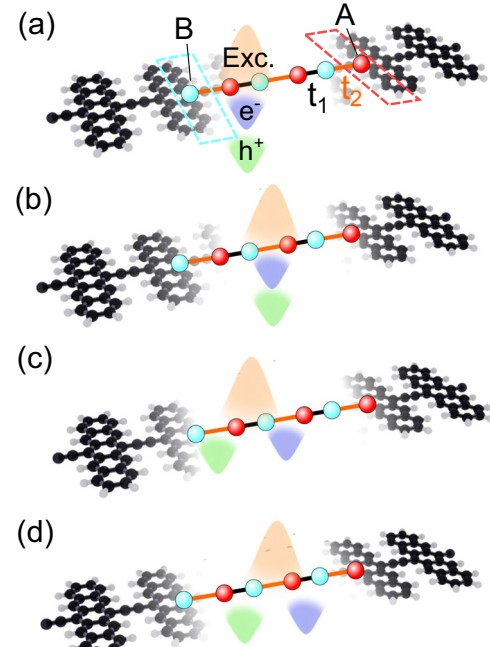

**Fig. 3 | Topological excitons in real space.** Real space realisation of different combinations of Wannier centre shifts with respect to sublattice sites $A$, $B$ admitting electron hopping amplitudes $t_1$, $t_2$. The Wannier centre shifts correspond to the Berry phases of electrons ($e^-$), holes ($h^+$), and excitons (Exc.) shown in blue, green, and orange, respectively. The interaction-driven configurations of the formation of topological and trivial excitons are shown schematically. **a** Trivial electrons and holes forming trivial excitons ($P_{exc} = 0$) in few-ring polyacenes. **b** Topological electrons and holes forming topological excitons ($P_{exc} = 1$) in many-ring polyacenes. The other possibilities include: **c** Topological electrons and holes forming trivial excitons ($P_{exc} = 0$). **d** Topological excitons ($P_{exc} = 1$) formed out of trivial electrons and holes, as conjectured in ref. 33. We find that only cases (**a**, **b**) can be realised in the excitonic bands of polyacenes.

short-range interactions with $i = 1, 2$. The electronic hopping terms become negligible and the nearest-neighbour interactions, modelled with the Hubbard terms, take over the role of the quasiparticle hoppings, with density operators taking the role of creation and annihilation operators after a particle-hole transformation $h^\dagger_{A/B} = c_{A/B}$, and a bosonisation to the localised exciton basis $b^\dagger_{A/B} = c^\dagger_{A/B} h^\dagger_{A/B}$, see Supplementary Information (SI). On relabelling the dominant nearest-neighbour interaction energies $U_1$ and $U_2$ as $2t_1^{exc}$ and $2t_2^{exc}$, the dual Hamiltonian takes an SSH form:

$$H = -t_1^{exc} \sum_j b^\dagger_{B,j} b_{A,j} - t_2^{exc} \sum_j b^\dagger_{B+1,j} b_{A,j} + \text{h.c.}. \quad (6)$$

Interestingly, this regime with dominant interactions described by the dual Hamiltonian in Eq. (6) hosts two new phases which are not present in the phase diagram in Fig. 2. When $t_1^{exc} > t_2^{exc}$ and $t_2 \gg t_1$, we obtain a regime in which we have topological electrons and holes with $\phi_e = \phi_h = \pi$ but trivial excitons with $\phi_{exc} = 0$. This regime is associated with electron Wannier centres localised at the edges of the unit cells but exciton Wannier centres localised at the centres of the unit cells, see Fig. 3c. When $t_2^{exc} > t_1^{exc}$ and $t_1 \gg t_2$, we obtain a regime in which we have trivial electrons and holes with $\phi_e = \phi_h = 0$ but topological excitons with $\phi_{exc} = \pi$. This regime, which was recently described in ref. 33, is associated with electron Wannier centres localised at the centres of the unit cells but exciton Wannier centres localised at the edges of the unit cells, see Fig. 3d. Both of these regimes exhibit exciton phases in which the topology is either driven or suppressed by the electron-hole interactions rather than by the underlying single-particle states.

A natural question to ask at this point is why the two interaction-dominated regimes realisable in the SSH-Hubbard model[33] are missing from Fig. 2. To understand this, we note that in the SSH-Hubbard model[33] the single-particle parameters $t_1$ and $t_2$ and the interaction parameters $U_1$ and $U_2$ are taken as independent parameters. However, in a more realistic setting, such as that provided by the Wannier equation used to build the phase diagram in Fig. 2, they are not independent. As such, our numerical results obtained from the Wannier equation show that, for the one-dimensional systems studied, excitons only exhibit topology inherited from the underlying single-particle electron and hole topologies. An interesting question to explore in future work would be the possibility of interaction-driven topology in excitons, perhaps in higher-dimensional settings.

## Topological excitons in polyacenes

Most importantly, we identify a family of organic acene one-dimensional polymers[34,35] as candidate materials to realise the predicted exciton topological phase described above. These materials are formed of $N$-ring acene molecules linked by a carbon-carbon bond on the central carbon atoms (see Fig. 3 for a schematic), and we consider the cases $N = 3, 5, 7$.

We use density functional theory[36–39] to evaluate the single-particle states of these polyacenes, and then fit the results to the SSH model to identify effective $t_1$ and $t_2$ hopping parameters. We indicate the location of polyanthracene ($N = 3$), polypentacene ($N = 5$) and polyheptacene ($N = 7$) in the phase diagram of Fig. 2a, providing material realisations of both trivial and topological excitonic phases.

The excitonic topology of polyacenes can be controlled through the application of strain along the polymer chains, as illustrated in Fig. 2b. Unstrained polyanthracene sits in region I, and by applying a tensile (+5%) or compressive (−5%) strain the bandstructure can be made less or more dispersive, respectively. In the case of polypentacene, which sits in region II for the unstrained case, strain has the opposite effect on the evolution of the band dispersion due to the topological electronic band inversion with respect to the polyanthracene case. Interestingly, unstrained polypentacene sits sufficiently close to the trivial-topological phase boundary that tensile strain could potentially drive a topological phase transition in this compound. From our density functional theory (DFT) calculations (see Methods), we find that this transition occurs between +5% and +10% strain, demonstrating that the topology of excitons can be controlled via strain, see also Supplementary Fig. 3. More generally, it would be interesting to explore how topology can be manipulated by other external parameters besides strain, for example through temperature or external electromagnetic fields[40].

## Excitonic quantum geometry

The excitonic topology described above has a direct implication on a geometric property, the spread (variance) of the excitonic states in the centre-of-mass coordinate **R**. To understand this relation, we consider the quantum metric $g_{ij}^{\text{exc}}$ [41,42], which is a tensor made of symmetrised derivatives of momentum-space Bloch states, and defined as:

$$g_{ij}^{\text{exc}} = \frac{1}{2}\left[\left\langle \partial_{Q_i} u_{\mathbf{Q}}^{\text{exc}} | 1 - \hat{P} | \partial_{Q_j} u_{\mathbf{Q}}^{\text{exc}} \right\rangle + \text{c.c.}\right], \quad (7)$$

where $\hat{P} = |u_{\mathbf{Q}}^{\text{exc}}\rangle\langle u_{\mathbf{Q}}^{\text{exc}}|$ is a projector onto the excitonic band of interest. More generally, it can be considered to be the real part of a Hermitian quantum geometric tensor (QGT)[41,43], which can be written as[41,44]:

$$Q_{ij}^{\text{exc}} = \left\langle \partial_{Q_i} u_{\mathbf{Q}}^{\text{exc}} | 1 - \hat{P} | \partial_{Q_j} u_{\mathbf{Q}}^{\text{exc}} \right\rangle. \quad (8)$$

The imaginary part of the QGT corresponds to the Berry curvature, encoding the topology that underlies quantum Hall phenomena, while the positive-semidefiniteness of the QGT equips the metric and the Berry curvature with a set of physical bounds related to optics, superconductivity, and quantum transport[43]. As we show in the following, the QGT directly determines the spread of the excitonic states. The geometric meaning of its real part, the metric, can be further identified by defining a Fubini-Study metric[41], $ds^2 = 1 - |\langle u_{\mathbf{Q}}^{\text{exc}} | u_{\mathbf{Q}+d\mathbf{Q}}^{\text{exc}}\rangle|^2$, and recognizing that:

$$ds^2 = g_{ij}^{\text{exc}} dQ^i dQ^j, \quad (9)$$

which is reminiscent of the well-known relation between the metric and the spacetime intervals in general relativity, and where the Einstein summation convention is implicitly assumed.

Focusing on one spatial dimension, which we refer to as the $x$ direction, the metric has a single component $g_{xx}^{\text{exc}}$. The metric is related to the exciton variance $\xi^2 \equiv \text{Var}\, R = \langle R^2 \rangle - \langle R \rangle^2$ of the centre of the maximally-localised exciton Wannier functions (MLXWF)[45] in real space. Explicitly, for a unit cell of length $a$, we have $\xi^2 = \frac{a}{2\pi} \int_{\text{BZ}} dQ\, g_{xx}^{\text{exc}}$. Using a Cauchy-Schwarz type inequality, we find (see Methods):

$$\xi^2 \geq \frac{a^2 P_{\text{exc}}^2}{4}. \quad (10)$$

This relation explicitly shows that the size of topological excitons, as captured by the spread $\xi^2$ of the excitonic Wannier functions, should be comparable to, or exceed, the size of the unit cell. That is, topological excitons with $P_{\text{exc}} = 1$ are shifted to the boundary of the unit cell $\langle R \rangle = a/2$ and their spread $\xi^2$ is bounded from below, resulting in the centre-of-mass $R$ of an exciton being localised within a characteristic length $\xi \sim a/2$. In the presence of a localised electron or hole (e.g. through defect pinning), the bound on the delocalisation of the centre-of-mass $R$ position also determines the effective size of the exciton characterised by the electron-hole distance $r$. This is consistent with the observed transition from localised "Frenkel" to spread out "Mott-Wannier" excitons in polyacenes[35].

To illustrate the theoretical results on the lower bound on the exciton centre-of-mass spread, we numerically demonstrate the spread of the excitons in Fig. 4. We confirm that whenever the topological excitonic invariant $P_{\text{exc}}$ is non-trivial, the diameter associated with the variance of the real space exciton wavefunctions exceeds the size of the unit cell. By contrast, Fig. 4 also shows that for an excitonic inversion invariant $P_{\text{exc}} = 0$, the excitons have no lower bound on their spread and they are more localised compared to their topological counterparts.

Finally, we find that increasing the dielectric screening can be used to fine tune the exciton spread, while satisfying the topological bound. In an experimental setting, the screening strength can be controlled through substrate choice, but also through chemical functionalisation. More details can be found in the SI.

## Higher-energy excitonic bands

The discussion up to this point has focused on the lowest-energy excitonic band. This is typically the most interesting excitonic band from an optoelectronics point of view, but the discussion above applies equally to higher energy excitonic bands. To illustrate this, Fig. 5a–d illustrates the exciton envelope functions for excitonic bands $n = 1, 2, 3, 4$ while the associated spread of the exciton Wannier states are shown in Fig. 5e–h. The topological phase diagram for each of these states is the same owing to the vanishing envelope contribution to the excitonic Berry connection[25].

The quantum geometric bound driven by the excitonic band topology persists in the higher excitonic bands, see Fig. 5e–h. In the higher bands, quantum geometry takes a more dominant role, even without the presence of non-trivial topology, due to the broader envelope functions of the weaker-bound higher energy excitonic states (see further discussion of this point in SI).

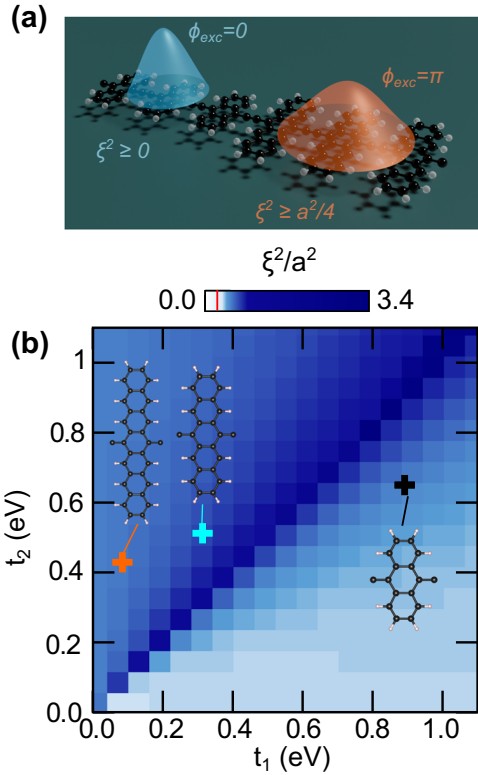

**Fig. 4 | Quantum geometry of topological excitons. a** Illustration of topological (orange) and trivial (blue) excitons in polyacenes with the corresponding excitonic Berry phases and excitonic Wannier state spreads bounded below by the excitonic quantum metric. **b** Spread of the maximally-localised excitonic Wannier states $\xi^2$ as a function of the single-particle hopping parameters $t_1$ and $t_2$, with the excitonic phase bounded by the excitonic quantum geometry (red line in gradient bar). The corresponding parameters and $\xi^2$ values for polyheptacene (orange cross), poly-pentacene (cyan cross), and polyanthracene (black cross) are indicated.

## Discussion

Our findings open up multiple research avenues. From a theoretical point of view, the SSH-like topology discussed above, also referred to as obstructed insulators, is the simplest kind of band topology. Extending the study of topological excitons to higher dimensional systems and to materials with different types of (crystalline) symmetries is expected to greatly expand excitonic topology and quantum geometry[24–26,33,44]. Another interesting research direction is to extend the discussion above to spinful models, which would enable a distinction between topological properties of singlet and triplet excitons, which in this work we find to be qualitatively the same, in a broader range of materials.

These potential theoretical extensions would naturally fit with active areas of experimental exciton research. Excitons are experimentally studied through their optical, dynamics, and transport properties, using techniques such as pump-probe experiments. The findings reported above already anticipate some potential experimental manifestations of excitonic topology. First, the strain-controllable exciton topologies and geometries exhibited by different exciton states (see Figs. 2 and 5) suggest that optical probes could selectively target topologically-distinct excitons under different external conditions applied to the same material, as well as unusual dynamics as these excitons relax towards the lowest energy states following photoexcitation. Second, the lower bound set on the exciton size by quantum geometry (see Fig. 4) suggests that transport will be enhanced for topological excitons compared to their trivial counterparts, an intriguing possibility we have very recently explored in ref. 40. Third, a hallmark of topology is the so-called bulk-boundary

correspondence, in which the bulk topology is associated with unusual boundary states that are often protected against scattering and support dissipationless transport in higher dimensions. In the case of topological excitons, such topological excitonic boundary states should be experimentally observable in local optical conductivity measurements, as suggested by ref. 33.

From a materials point of view, the interplay between topology, excitons, and organic polyacenes reported above is only the starting point. Organic semiconductors provide a more general platform to explore excitonic topology, with organic semiconductors in one, two, and three dimensions often exhibiting strongly bound excitons, due to their weak dielectric screening. Additionally, it should be possible to harness the decades of experience in organic chemistry to manipulate organic semiconductors to explore many potential different exciton topological regimes, such as those described in Fig. 3, but notably also richer topologies in higher dimensions. Beyond organic semi-conductors, inorganic two-dimensional materials[26,46] and van der Waals bonded layered materials[47–49] also exhibit strongly bound excitons, providing additional material platforms to explore excitonic (crystalline) topology.

Overall, we establish a connection between topological physics, common in the study of electronic phases, and the field of excitons in organic semiconductors. We describe a simple proof-of-principle for excitonic topology in the form of one dimensional crystalline topological invariants, constructing a full excitonic topological phase diagram with different regimes reflecting the interplay between the underlying (so-called obstructed) electron and hole topologies with the new excitonic topology. We also present a family of organic polymers that host the predicted exciton topologies, and demonstrate the manipulation of this topology by applying experimentally realisable strains. Finally, we discover that excitonic topology is related to the spatial localisation of excitons, as determined by Riemannian metric identities. In particular, we find a lower bound on the exciton size for topologically non-trivial excitons. We show that the associated, topologically-bounded, excitonic quantum geometry can be further controlled with dielectric screening through a substrate choice, see Supplementary Fig. 2. Our results set a benchmark for a potentially rich exploration of topological excitons in organic semiconductors and beyond, which has the potential to impact properties ranging from optical to extraordinary transport signatures.

## Methods

### First principles calculations

We perform density functional theory calculations using the QUANTUM ESPRESSO package[38,39] to study a family of polyacene chains. We use kinetic energy cutoffs of 80 Ry and 500 Ry for the wavefunction and charge density, respectively, and 12 $k$-points to sample the Brillouin zone along the periodic direction. We use GGA (PBE) norm-conserving pseudopotentials which were generated using the code ONCVPSP (Optimized Norm-Conserving Vanderbilt PSeudoPotential)[50] and can be found online via the Schlipf-Gygi norm-conserving pseudopotential library[51]. We impose a vacuum spacing of 34.3 Å in the planar direction perpendicular to the polymer chain, and a vacuum spacing of 27.52 Å in the out-of-plane direction, to minimise interactions between periodic images of the polyacene chains. We perform a structural optimisation of the internal atomic coordinates to reduce the forces below 0.0015 Ry/Å. To model the impact of strain we expand/compress the unit cell in the polymer chain direction before relaxing the atomic positions. For further details concerning atomic positions, see Supplementary Data 1.

We calculate the band structures of several polyacenes, as shown in Fig. 6. The hopping parameters of the SSH tight-binding model were deduced from these calculations. The SSH sites, $A$ and $B$, on the molecular unit cell are illustrated with the red and blue boxes in Fig. 6d, with the $t_1$ hopping across an acene molecule and the $t_2$ hopping across the carbon link between acenes[34].

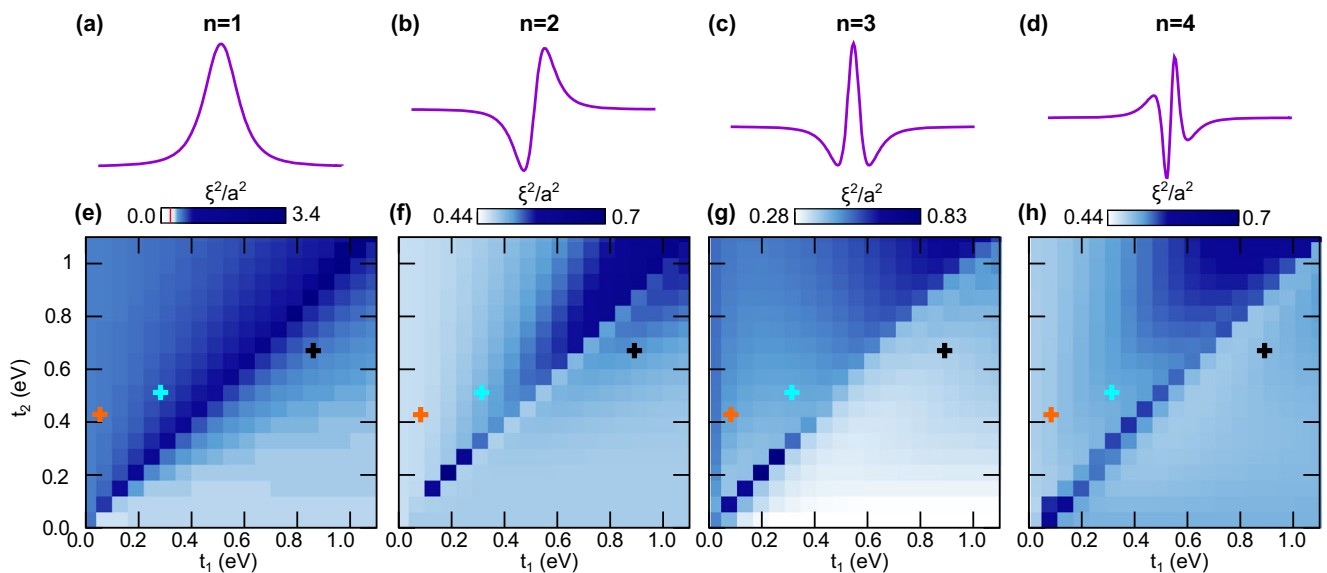

**Fig. 5 | Topological excitons in higher excitonic bands. a–d** Envelope functions $\psi_{\mathbf{Q}=0}(\mathbf{k})$ for different excitonic bands $n$ = 1, 2, 3, 4, respectively. **e–h** Spread of the excitonic Wannier states $\xi^2$, with the excitonic topological phases bounded by the excitonic quantum geometry (red line in gradient bar). Although the numerical results match the general theory and topological phase diagram intimately, we note that in higher excitonic bands $n$ = 3, 4 (**g, h**), there is a larger spread $\xi^2$ than in the topologically trivial excitonic wavefunctions of lower bands (**e, f**), due to the contributions of more delocalised envelope function (see also SI). The excitons in polypentacene (cyan cross), polyheptacene (black cross) and polyheptacene (orange cross) realise different excitonic topologies.

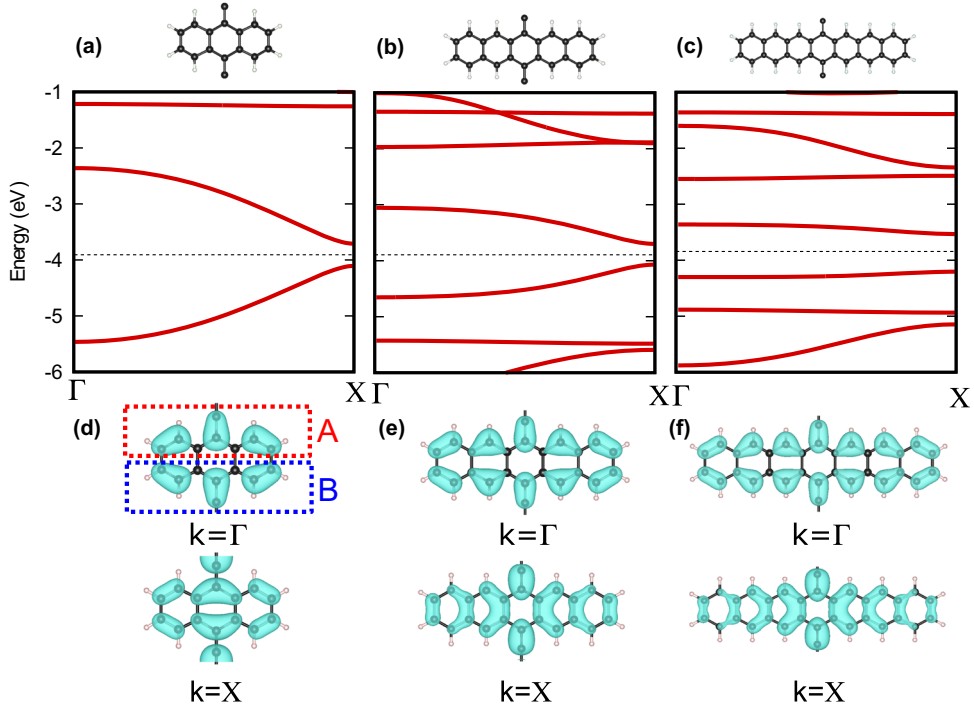

**Fig. 6 | Electronic band topology in organic semiconductors.** Electronic bands in **a** polyanthracene, **b** polypentacene, and **c** polyheptacene, calculated using DFT. The corresponding Fermi levels are indicated with black dashed lines. **d–f** Plots of the electronic wavefunctions for the valence band in the polyacenes, which is contributed by the $p_z$-orbitals. The shift of the charge centre, as well as the change of the parity of the wavefunction, as indicated by the inversion symmetry eigenvalues at the high symmetry points of a one-dimensional momentum space, $k = \Gamma$ (BZ centre), and $k = X$ (BZ edge), can be directly observed (on comparing $P_{\mathrm{exc}} = 0$ to $P_{\mathrm{exc}} = 1$) in the DFT results. The approximate $A$ and $B$ sublattice sites in the molecular unit cells are marked with the red and blue boxes, respectively.

Using these calculations, we identify polyanthracene ($N$ = 3) as exhibiting topologically trivial electronic states, while polypentacene ($N$ = 5) and polyheptacene ($N$ = 7) exhibit topological electronic states. Repeating the same analysis using the many-body $GW$ approximation to calculate the band structure of the polyacenes leads to the same

topological transition as the polyacene chain length increases as that calculated at the DFT level, but with the transition occurring for longer chain lengths[35]. Specifically, the transition from trivial to topological electrons and holes occurs between the polypentacene ($N$ = 5) and polyheptacene ($N$ = 7) polymers, in contrast with our and previous DFT

calculations and experiments[34] where the transition occurs between $N = 3$ and $N = 5$.

## Wannier equation

We obtain the excitonic wavefunctions and associated bands by combining the electronic and hole states from the SSH model or from the first principles DFT calculations, with the envelope part of the exciton wavefunction obtained by solving the Wannier equation. For the illustrations of the envelope part solutions, see also Supplementary Fig. 1.

For each pair $(t_1, t_2)$, we solve the Wannier equation:

$$\sum_{k'} h_{k,k'}(Q)\psi_Q^n(k) = E_n(Q)\psi_Q^n(k), \tag{11}$$

where $\psi_Q^n(k)$ is the envelope function of the $n$th excitonic band. In this expression, we have:

$$h_{k,k'}(Q) = (E_{k+Q/2}^e - E_{k-Q/2}^h)\delta_{k,k'} - W_{k,-k',Q}, \tag{12}$$

with electron/hole band energies $E^{e(h)} = E_{+(-)}$, and the interaction matrix defined as[26]:

$$
\begin{aligned}
W_{k,-k',Q} &= V_{NR}(k-k') \\
&\times \sum_{i,j\in\{A,B\}} \varphi_{i,k+Q/2}^* \varphi_{j,-k'-Q/2}^* \varphi_{j,k-Q/2} \varphi_{i,k'+Q/2},
\end{aligned}
$$

where $\varphi_{i,k}$ is the wavefunction amplitude from the SSH model on site $i$ with momentum $k$, and $V_{NR}$ is the Coulomb potential for a one-dimensional system that describes the dielectric screening. We show the excitonic band dispersions for the lowest four excitonic bands of different polyacenes in Fig. 7.

The Wannier equation is highly dependent on the dielectric screening, where an accurate screening model is paramount to adequately capture the excitonic physics. In the polyacene chains, the Coulomb potential corresponds to that of one-dimensional nanoribbons. Hence, consistently with ref. 52, on solving Poisson's equation, we arrive at the following form of the Coulomb potential:

$$V_{NR}(Q) = \frac{e_0^2}{4\pi\epsilon_0} \frac{K_0(QL/2)}{\varepsilon_s + 8Q^2\alpha_{1D}K_0(QL/2)}, \tag{13}$$

where $L$ is the width of the ribbon, $\alpha_{1D}$ is the screening parameter (polarisability per unit length), $\varepsilon_s$ is the background screening, and $K_0$ is a modified Bessel function of the second kind. We take $L$ to be the lateral size of the polyacene ribbon and fix $\alpha_{1D} = 0.05$ nm$^{-2}$ following previous work[52]. For small $Q$, corresponding to long range interactions, the screening is dominated by the environment as determined by $V_{NR}(Q \to 0) \sim -\log(QL/2)/\varepsilon_s$. At large $Q$, corresponding to short distances, the screening is approximately $V_{NR}(Q \to 0) \sim 1/(\alpha_{1D}Q^2)$, resembling the Coulomb interaction in bulk systems. The background screening $\varepsilon_s$, defined as the average of the dielectric above and below the organic layer, further modulates the Coulomb interaction. We consider common background dielectrics in this work: vacuum ($\varepsilon_s = 1$), SiO$_2$ substrate ($\varepsilon_s = 2.45$), and hBN encapsulation ($\varepsilon_s = 4.5$)[53].

## Numerical excitonic Berry phases

To evaluate the excitonic Berry phases, we use a discretisation of the Brillouin zone $Q = [Q(1), Q(2), \ldots, Q(N_Q)]$, where we choose $Q(1) = 0$ and $Q(N_Q) = 2\pi/a - \Delta Q$, with lattice parameter $a$ and grid spacing $\Delta Q = 2\pi/(N_Q a)$. We use the parallel-transport gauge to numerically evaluate the Berry phase given by Eq. (2), following ref. 54:

$$\phi_{exc} = -\Im \log\left[\langle u_{Q(1)}^{exc}|u_{Q(2)}^{exc}\rangle \cdots \langle u_{Q(N_Q-1)}^{exc}|u_{Q(N_Q)}^{exc}\rangle\right]. \tag{14}$$

We use a $Q$-grid of size $N_Q = 400$, with convergence within a numerical error of $\Delta\phi_{exc} \sim 10^{-3}$ for the excitonic Berry phase obtained above $N_Q = 200$.

## SSH-Hubbard model

In this Section we expand on the effective model discussed in the main text. We begin by building an electronic effective theoretical model based on the one-dimensional SSH Hamiltonian[31,32]. We write Eq. (5) of the main text in the momentum-space Bloch orbital basis by introducing $c_{Ak}^\dagger / c_{Bk}$, on Fourier transforming the electronic creation and annihilation operators in the position basis $c_{Ak}^\dagger = \frac{1}{\sqrt{L}}\sum_{j=1} e^{ik(r_e)_j} c_{A,j}^\dagger$,

$$H = \sum_{k;\alpha,\beta=A,B} \left[H^{SSH}(k)\right]_{\alpha\beta} c_{\alpha k}^\dagger c_{\beta k}, \tag{15}$$

with the electron position operator $(r_e)_j = ja$. In this basis, we have an effective Bloch Hamiltonian matrix for low energy, i.e. top valence and

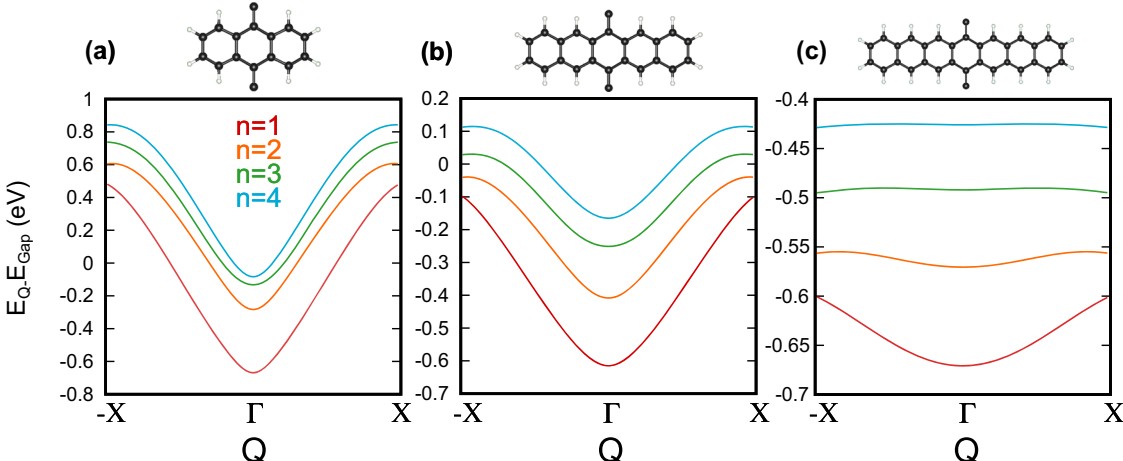

**Fig. 7 | Excitonic bands in organic polymers.** Excitonic bands obtained from DFT single-particle states and the solution of the Wannier equation for the envelope function. We show the dispersion of the excitonic bands $n = 1$ (red), $n = 2$ (orange), $n = 3$ (green), and $n = 4$ (blue) of **a** polyanthracene ($N = 3$), **b** polypentacene ($N = 5$), and **c** polyheptacene ($N = 7$). The excitonic bands are manifestly inversion-symmetric, while the excitonic bandwidths and dispersion decrease with increasing number of rings $N$ in the monomers. We also note that as the excitonic bands approach the zero energy level, they merge into the continuum of electronic energies.

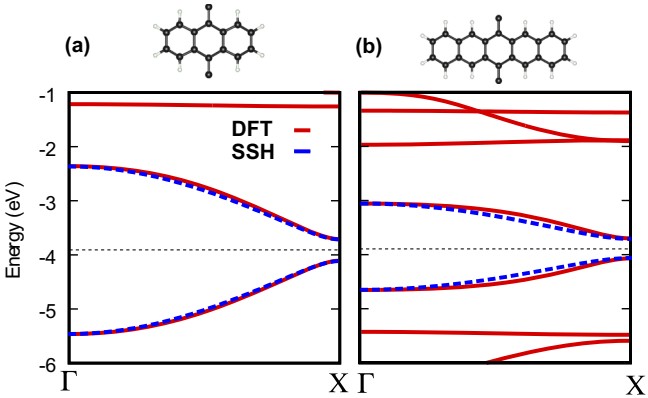

**Fig. 8 | DFT bands and the electronic SSH model.** Low energy band fitting of the effective SSH model (blue) to the band structure obtained with DFT calculations (red) for **a** polyanthracene and **b** polypentacene. The black dashed lines indicate the Fermi levels. We note that apart from the energy fitting, the form of the electronic wavefunction and its parity, as provided in Fig. 6 and consistent with ref. 34, are necessary to show the topological nature of the electronic states that further contribute to the formation of topological excitons.

bottom conduction, bands, which reads:

$$H^{\mathrm{SSH}}(k) = \mathbf{d}(k) \cdot \boldsymbol{\sigma}, \qquad (16)$$

where $\mathbf{d}(k) = (t_1 + t_2 \cos k, t_2 \sin k, 0)^T$, with $\boldsymbol{\sigma} = (\sigma_x, \sigma_y, \sigma_z)$, the vector of Pauli matrices. The hopping parameters $t_1$ and $t_2$ can be retrieved to describe specific polyacenes by fitting them to the associated first principles calculations after Wannierisation. In this electronic model, the wavefunctions of electrons and holes take the compact form $|u_k^{\mathrm{e/h}}\rangle = \frac{1}{\sqrt{2|\mathbf{d}|^2}}(d_x \pm i d_y, |d|)^T$. Correspondingly, we further demonstrate the quantitative fit of the discussed electronic SSH model to the first principles band structures; see Fig. 8 below.

Having set up the electronic problem, we now elaborate on the theoretical model for the phenomenology of the interaction-dependent excitonic topology. We start with a SSH-Hubbard-type model with up to the $n$th neighbour interactions $U_i$, with $i = 1, 2, 3, \dots, n$ in real space:

$$
\begin{aligned}
H = & \sum_j \epsilon_A n_{A,j} + \sum_j \epsilon_B n_{B,j} \\
& - t_1 \sum_j c_{A,j}^\dagger c_{B,j} + \mathrm{h.c.} - t_2 \sum_j c_{A,j+1}^\dagger c_{B,j} + \mathrm{h.c.} \\
& + \sum_j U_A n_{A,j}(n_{A,j} - 1) + \sum_j U_B n_{B,j}(n_{B,j} - 1) \\
& + \frac{1}{2} \sum_j \sum_{\alpha,\beta=A,B} \sum_{i=1}^n \left[ U_{2i-1}^{\alpha\beta} n_{\alpha,j-i} n_{\beta,j} + U_{2i}^{\alpha\beta} n_{\alpha,j+i} n_{\beta,j} \right],
\end{aligned}
\qquad (17)
$$

where the density operator is $n_{A,j} = c_{A,j}^\dagger c_{A,j}$. We set the onsite energies to zero, $\epsilon_A = \epsilon_B = 0$, and the onsite interaction terms of strength $U_A$, $U_B$ are further set to vanish within a spinless model. Consistently, the model admits no self-interactions. We additionally note that if the screened electron-electron interaction potential takes a Coulomb-like decaying form, $V(r - r') \sim \frac{1}{\varepsilon_s |r-r'|}$, the interaction strengths can be completely suppressed as $U_1, U_2 \sim 1/\varepsilon_s \to 0$, when formally, $\varepsilon_s \to \infty$. Equivalently, this limit yields a hard-core repulsive potential $V(r - r') \sim \delta(r - r')$, where $\delta(r - r')$ is the Dirac delta function.

In the real material context, and consistently with our fitting, we stress that all the phenomenological model parameters $t_1$, $t_2$, $U_i$, etc., are self-contained in the self-consistent solution of the Wannier equation, but otherwise these can be directly estimated to an arbitrary

order from the electronic Wannier functions. For the standard computation of the Wannier functions using WANNIER90, see ref. 55.

## Data availability
All datasets for the plots of this study are available upon request to the authors.

## Code availability
All codes and associated data are reproducible with information in the manuscript. All first-principles calculation input files are available upon request to the authors.

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

## Acknowledgements

The authors thank Richard Friend, Akshay Rao, Sun-Woo Kim, Gaurav Chaudhary, Arjun Ashoka, Henry Davenport, and Frank Schindler for helpful discussions. This project was supported by funding from the Rod Smallwood Studentship at Trinity College, Cambridge (W.J.J.). We acknowledge support from a EPSRC Programme grant EP/W017091/1 (J.J.P.T., and B.M.), as well as from UKRI Future Leaders Fellowship MR/V023926/1, from the Gianna Angelopoulos Programme for Science, Technology, and Innovation, and from the Winton Programme for the Physics of Sustainability (B.M.). We acknowledge funding from a New Investigator Award, EPSRC grant EP/W00187X/1, a EPSRC ERC underwrite grant EP/X025829/1, and a Royal Society exchange grant IES/R1/221060, as well as from Trinity College, Cambridge (R.-J.S.).

## Author contributions

B.M. and R.-J.S. initiated the project. W.J.J. performed initial theoretical analysis of excitonic topology and geometry with inputs from R.-J.S. J.J.P.T. performed all numerical first-principles calculations with inputs from B.M. W.J.J. and R.-J.S. constructed theoretical phenomenological model with inputs from J.J.P.T. and B.M. All authors discussed the results and substantially contributed to the writing of the manuscript. The final form of the manuscript, including Methods and Supplementary Information, benefitted from input from all authors.

## Competing interests

The authors declare no competing interests.
