## [Transparent Peer Review file · Nature Communications]

Excitonic topology and quantum geometry in organic semiconductors

Corresponding Author: Professor Robert-Jan Slager

Version 0:

Reviewer comments:

Reviewer #1

(Remarks to the Author)

The article introduces the concept of excitonic topology in organic semiconductors, indicating that excitons can exhibit topologically non-trivial states. The study identifies a one-dimensional polyacene family of organic semiconductors demonstrating the predicted excitonic topological phases. By leveraging quantum Riemannian geometry, the paper predicts that topologically non-trivial excitons have a lower bound on their spatial spread, which can be significantly larger than the size of a unit cell.

While reading the article, the following questions arose. Unfortunately, we cannot recommend the article for publication in Nature Communications. We think the manuscript would merit publication in a less rigorous scientific journal if it resolves the questions below.

1. Model validation:

The Su–Schrieffer–Heeger (SSH) model is one of the basic models in describing band topology in condensed matter systems (see, e.g., Refs. [1-5]). Furthermore, various extensions of the SSH model are used to study additional aspects of topological physics [6-10]. The authors do not mention this important fact. Meanwhile, most of the topologically non-trivial properties discussed in the paper are related to it.

Consequently, the theoretical framework proposed in the study needs to be validated, which is essential to confirm the feasibility and practical relevance of the findings. To be concrete, why can the SSH model describe the single-particle electron and hole states of polyacene (Eq. 5) without needing to consider the geometry of polyacene?

2. Controllability of Excitonic Topology:

The controllability of excitonic topology through chemical functionalization and dielectric environment is highlighted in the abstract. A more in-depth analysis of the mechanisms and techniques for achieving this control in the main part would enhance the practical relevance of the research.

3. Clarification on Figure 3:

Is Figure 3 a schematic diagram only, or is it numerically computed? If it is numerically computed, what parameters are used in each subfigure?

4. Benefits of Introducing Quantum Geometry:

What benefits do we get from introducing quantum geometry (the imaginary part of which recovers Berry curvature)? Figure 2a and Figure 4b seem to indicate that they carry the same information. Is the quantum geometric tensor more experimentally accessible through the spatial spread of the states, for example?

5. Lower and Upper Bounds of Excitonic Spread:

Equation 10 gives a nontrivial lower bound for the topologically nontrivial state. Is there any upper bound for the topologically trivial state? If there is no bound, can a trivial state also have a large spread?

6. From the materials point of view, the spin of the exciton is one of the most important characteristics defining the most

important optical properties of organic semiconductors. In this connection, it is important to include the spin into consideration. Unfortunately, from the current study, it is not intuitive whether including the spin will preserve the band topology, particularly the part related to the Wannier envelope of the wave function. The authors mention the relevance of the excitonic spin in the opening paragraph of the Discussion section, but this important issue must be discussed more thoroughly.

References

- [1] M. Atala et al., Direct measurement of the Zak phase in topological Bloch bands. *Nat. Phys.* 9, 795–800 (2013).
- [2] L. Wang, M. Troyer, and X. Dai, Topological charge pumping in a one-dimensional optical lattice. *Phys. Rev. Lett.* 111, 026802 (2013).
- [3] M. Lohse, C. Schweizer, O. Zilberberg, M. Aidelsburger, and I. Bloch, A Thouless quantum pump with ultracold bosonic atoms in an optical superlattice. *Nat. Phys.* 12, 350 (2016).
- [4] S. Nakajima et al., Topological Thouless pumping of ultracold fermions. *Nat. Phys.* 12, 296 (2016).
- [5] M. Leder et al., Real-space imaging of a topologically protected edge state with ultracold atoms in an amplitude-chirped optical lattice. *Nat. Comm.* 7, 13112 (2016).
- [6] Y.-G. Peng et al., Experimental demonstration of anomalous Floquet topological insulator for sound. *Nat. Comm.* 7, 13368 (2016).
- [7] A. Gómez-León and G. Platero, Floquet-bloch theory and topology in periodically driven lattices. *Phys. Rev. Lett.* 110, 200403 (2013).
- [8] V. Dal Lago, M. Atala, and L. E. F. F. Torres, Floquet topological transitions in a driven one-dimensional topological insulator. *Phys. Rev. A* 92, 023624 (2015).
- [9] F. A. An, E. J. Meier, and B. Gadway, Engineering a flux-dependent mobility edge in disordered zigzag chains. *Phys. Rev. X* 8, 031045 (2018).
- [10] B. Perez-Gonzalez, M. Bello, A. Gomez-Leon, and G. Platero, Interplay between long-range hopping and disorder in topological systems. *Phys. Rev. B* 99, 035146 (2019).

Reviewer #2

(Remarks to the Author)

The authors perform DFT calculations on various polyenes and model their electronic properties with a tight-binding approach characterized by two hopping per unit cell t_1 and t_2 . Additionally, they obtain the electronic orbitals essential for determining the excitonic bands of the system by solving the Wannier equation. Once the excitonic band structure is obtained the manuscript explores the topology and geometric effects of excitonic bands

I have some doubts and questions on the results that the authors should clarify before publication.

1) In Fig.2, the exciton energies are the same for the pair $\{t_1, t_2\}$ as they are for the pair $\{t_2, t_1\}$?

2) I don't understand why does not appear a phase IV, similar to phase III. In Fig.2

The exciton Hamiltonian possess spatial inversion symmetry. Due to this symmetry and the translational symmetry of the system, there are two inequivalent inversion centers: one at $x=0$ and another at $x=a/2 \pmod{a}$. These correspond to the two possible locations of the exciton Wannier function center, as discussed by Zak (PRL 45, 1025). The Wannier center at $x=0$ corresponds to an excitonic Zak (Berry) phase $\phi=0$, while the center at $x=a/2$ corresponds to $\phi=\pi$ (Zak PRL 48, 359), both modulo 2π .

In the studied system, the inversion centers are located at the centers of the bonds with hopping parameters t_1 and t_2 . If the origin of coordinates is shifted from the center of a bond with hopping t_1 to the center of a bond with hopping t_2 , the center of the excitonic Wannier function shifts by $a/2$. Consequently, the Zak phase changes by π .

Thus, if a system with hopping parameters $\{t_1, t_2\}$ has a Zak phase of 0 and a Wannier center at 0, interchanging the hopping parameters should lead to an energy degenerated Zak phase of π and a Wannier center at $a/2$.

Therefore, In this system, I would expect that the existence of a phase III should accompanied by the existence of a complementary phase IV.

I suspect that, in Fig.2, when the hopping amplitudes are interchanged the form and position of the electron and hole wavefunction are not interchanged. And therefore

This model are not exactly equivalent to the SSH model, where two degenerated topologically different phases appear.

If this is the case, which orbitals are used for each pair of $\{t_1, t_2\}$? DFT calculation only provide this information for a small number of polienes?

3) The Berry (Zak) phase is calculated using Eq. 14. The system has inversion symmetry. Is it possible to simplify this calculation by using the parity eigenvalues at the Time-Reversal Invariant Momenta (TRIMs), such as Γ and X? (Fu-Kane PRB 76 045302 2007).

4) Did you calculate the values of U1 and U2 at different points in the phase diagram, or did you estimate the values of U1 and U2 based on the form of the phase diagram?

5) How were the parameters t_1 and t_2 obtained from the DFT calculations? Did you use Wannier functions to derive these parameters, or did you simply fit the last two subbands using two hopping parameters? Plotting the tight-binding band structure over the DFT results could help assess the quality of the approximation.

6) In Fig. 7, you plot the excitonic bands for different organic semiconductors. In cases (b) and (c), it appears that the bands are degenerate at the X points. If this is the case, the bands are not isolated, and it would not be possible to compute the Berry phase for individual bands. Could you discuss on this?

7) The spread of the exciton Wannier function is a gauge-dependent quantity, as discussed in Vanderbilt's book. Which gauge did you use in your calculations? Did you try to minimize the size of the Wannier function?

8) I believe there is a typo, in the definition of $d(k)$ after Eq.20 in the Appendix. The sin and the cos seem to be interchanged.

9) In some sections the manuscript is difficult to follow. It is not clear which approximation is done in the calculations. Also, it is not clear if the discussion on the dualized model is based in numerical calculation. The similarities and differences with the SSH model are not clear. In order to follow the manuscript, it is necessary to frequently refer to the appendices. Therefore, it would be very helpful to include specific references to the relevant appendices in the main text whenever they are utilized.

The authors should revise the manuscript to clearly outline the approximations employed at each stage of their analysis. Additionally, they should discuss the connection with the SSH model.

Reviewer #3

(Remarks to the Author)

Reviewer #4

(Remarks to the Author)

The authors present a theoretical and numerical analysis of the topological properties of excitons in a family of one-dimensional systems based on the famous SSH model. This is a very timely topic which is being addressed only very recently by few groups and has the potential to push our understanding of excitons to higher grounds.

Having said that, I think the manuscript has room for necessary improvement:

1) Since, as the authors state: "The invariant P_{exc} precisely distinguishes two different topologies which are not transversable from one to another without closing a gap in the excitonic bands", an important issue regarding the transition from topologically trivial to non-trivial is the closing and opening of the gap, it would be important to see a more detailed analysis of the exciton bands. Now this analysis comes down to a single figure in the Methods section where nothing can actually be seen. In fact, according to the colours, no gap closing seems to occur from a) to b). What can be made out of this plot?

2) Region III in Fig. 2b,c, where the excitons become trivial on account of the interactions, takes over region II when interactions are quenched through a dielectric screening. I am afraid I cannot understand the arguments on page 4 that presumably explain this. Since the "trivialization" of the excitons originates in the interactions, Shouldn't region III shrink when the interactions are screened (weakened)? Is the change from long-range to short-range interactions responsible for

that? Does region II disappear altogether for a hard-core interaction model (presumably representing a very strong screening? Is the Hubbard model in the flat-band limit mimicking this case? In summary, I think the discussion should attempt to separate the effects of the range and the strength of the interaction in a clearer way and its connection to the very unclear discussion.

3) The final discussion contains too many promises, but no actual recipe for an experimental test of the topological properties described in the text. For instance, if an overall gap appears in the Brillouin zone one may expect localized excitons on the ends. However, I cannot imagine how this gap can appear judging from the bands presented. In my opinion, despite the very interesting theoretical insight gained, a likely or possible experimental manifestation of the predicted non-trivial topology would be necessary to make this work worth publishing in Nature Comms.

Reviewer #5

(Remarks to the Author)

Version 1:

Reviewer comments:

Reviewer #1

(Remarks to the Author)

The authors have improved the clarity of their manuscript. Specifically, they have clarified the role of the SSH model and expanded their discussion of the experimental relevance of strain-engineered excitons in the supplementary materials, etc.

However, a few concerns persist:

1 While the authors have clarified how the ring geometry is incorporated into the DFT calculation and demonstrated the fitting of the SSH model parameters to the DFT results, this raises another critical question: Is fitting energy alone sufficient to capture the topological nature of the state? Topological information is generally encoded not only in eigenenergies but also in wave functions, which demands further scrutiny.

2. The authors have addressed the experimental aspects.

3. No further comments.

4. and 5 are interconnected: Equation 10 establishes that the spread of topological excitons is lower bounded by $a^2/4 P_{\text{exc}}^2$, while Equation 3 defines P_{exc} as an integral of the Berry connection over the Brillouin zone. This implies that the quantum geometry tensor primarily acts as an intermediate quantity in the derivation, and its relevance to the paper's core findings remains unclear. Additionally, the experimental implications for diffusive transport and optical responses are undermined by the fact that even a trivial state could exhibit a large spread. Although the authors argue that the non-Abelian Berry connection determines the metric, so the large spread of trivial states is rare, without a more quantitative description, this assertion is weak, more or less.

6. No further comments.

7 An extra comment: The deferral of the full SSH-Hubbard Hamiltonian, which is used to calculate many results in the main text, to the supplementary materials undermines the manuscript's accessibility.

Reviewer #2

(Remarks to the Author)

I am satisfied with the changes made in the revised manuscript. I'm happy to recommend that the manuscript be published.

Reviewer #3

(Remarks to the Author)

Reviewer #4

(Remarks to the Author)

The authors have done a fair job answering all the questions and comments of the referees and the manuscript has been

considerably improved. The error spotted in the review process was a serious one, which considerably affected the results, but it has been amended and everything makes more sense now. I recommend publication.

Reviewer #5

(Remarks to the Author)

Version 2:

Reviewer comments:

Reviewer #1

(Remarks to the Author)

The authors have thoroughly addressed the questions and comments of the report, and the paper now deserves publication.

Reviewer #3

(Remarks to the Author)

Reviewer 1 (Remarks to the Author):

The article introduces the concept of excitonic topology in organic semiconductors, indicating that excitons can exhibit topologically non-trivial states. The study identifies a one-dimensional polyacene family of organic semiconductors demonstrating the predicted excitonic topological phases. By leveraging quantum Riemannian geometry, the paper predicts that topologically non-trivial excitons have a lower bound on their spatial spread, which can be significantly larger than the size of a unit cell.

We thank the Referee for taking their time to review our manuscript.

While reading the article, the following questions arose. Unfortunately, we cannot recommend the article for publication in Nature Communications. We think the manuscript would merit publication in a less rigorous scientific journal if it resolves the questions below.

We thank the Referees for their assessment of our work. Unfortunately, we can only disagree with their recommendation, as we detail point by point below, purely on the basis of scientific arguments. We are surprised by this negative recommendation, especially given the fact that most criticism is rather minor and does not challenge any of our results nor their groundbreaking nature, which was explicitly recognised by the other Referees.

Additionally, we would like to emphasise that we take pride in our work and are confident that all our results are within rigorous and timely conceptual settings. As such, we would never think of sending results to a “less rigorous” journal, and we do not think that it is collegiate of the Referees to make such a suggestion about our work.

1. Model validation:

The Su–Schrieffer–Heeger (SSH) model is one of the basic models in describing band topology in condensed matter systems (see, e.g., Refs. [1-5]). Furthermore, various extensions of the SSH model are used to study additional aspects of topological physics [6-10]. The authors do not mention this important fact. Meanwhile, most of the topologically non-trivial properties discussed in the paper are related to it.

We are rather puzzled by this comment. First, we agree that many extensions of the SSH model have been studied in literature. However, we do not understand how this fact impacts the evaluation of the correctness, novelty, and scientific interest of our work, as we explore a completely new phenomenon, namely *exciton* topology in one dimension (1D). Additionally, we emphasize that our work goes far beyond a purely phenomenological model study, as we use a full DFT analysis to propose 1D organics as hosts of the newly introduced non-trivial exciton topology.

While interesting, the provided references for free (non-interacting) fermion/boson generalisations of SSH chains (Refs. [7]–[10]) do not appear relevant for our work, especially the realisations of non-interacting Floquet physics in the SSH model (Refs. [7]–[8]), disorder (Ref. [10]), or in the case of other references (Refs. [3]–[4]), Thouless pumping. We would respectfully like to ask the Referees that they refrain from bringing up irrelevant references, the irrelevance of which we clarify in detail in a point-by-point manner below.

We stress that we *do* cite the foundational works by Su, Schrieffer, and Heeger [Phys. Rev. Lett. **42**, 1698], [Phys. Rev. B **22**, 2099], which introduce the electronic SSH model, whereas our findings for the topology of excitons in real materials, possible only under the presence of interactions, are unprecedented. The rest of the manuscript is built around the predicted excitonic topology protected under the inversion symmetry with DFT calculations in real materials. Here, the Referees should also notice that, contrary to the SSH model, which enjoys a chiral symmetry, the topological excitons generically do not have chiral symmetry, as manifested by the presented excitonic band structures. Instead, our topological excitons require solely crystalline inversion symmetry. Therefore not only the physics is different (we focus on the interacting problem of excitons hosting topology in real materials), but also, unlike for the electrons, the *non-interacting* SSH Hamiltonian cannot be generically fitted *for excitons* themselves, given these enjoy and demand fundamentally different combinations of symmetries. Within the dualisation argument (see Methods), we were rather clear that the SSH Hamiltonian *for excitons* can be only retrieved in the flat-band limit, which specifically admits the chiral symmetry. This concludes the argument that only the original references on SSH physics were relevant to the electronic and excitonic aspects of our work.

To make the puzzling nature of the requested references more explicit, below we summarise reference-by-reference why these works are irrelevant to our material, or in multiple cases, even to non-interacting SSH physics, despite the claims of the Referee:

Ref. [1] - M. Atala et al., Direct measurement of the Zak phase in topological Bloch bands. Nat. Phys. 9, 795–800 (2013).

This reference, as the authors write, reports a “*measurement of the geometric phase acquired by cold atoms moving in one-dimensional optical lattices*”. As the authors stress in the main text, “*the key idea is to combine coherent Bloch oscillations with Ramsey interferometry to determine the geometric Zak phase and reveal the underlying topological character of the Bloch bands*”. While the work cites the original work by Su-Schrieffer-Heeger (SSH) [Ref. (3)], the focus of the work is on cold-atom quantum simulation, rather than on *electronic* structure of a real organic material [unlike the original Ref. (3), which we **do** cite]. Therefore, this Reference is of no relevance to our manuscript, when it comes to introducing the (interacting) SSH model in a real material context, which is the key of our work. More importantly, all physics in this paper is non-interacting, which contrasts with the topological excitonic physics titular to our work. Notably, even in the future plans, the authors only consider non-interacting realisations: “*Making use of the recently demonstrated control of optical potentials at the single-site level, we plan to realise domain walls or sharp boundaries in the dimerized lattice that would allow us to directly study edge states and fractional charges for **non-interacting** fermions or hardcore bosons.*” The excitons titular to our work, require by definition interactions between electrons and holes, and therefore cannot be captured with a non-interacting model.

Ref. [2] - L. Wang, M. Troyer, and X. Dai, Topological charge pumping in a one-dimensional optical lattice. Phys. Rev. Lett. 111, 026802 (2013).

This reference is about proposing “*an experimental setup to realise topological charge pumping of cold fermionic atoms in a one-dimensional optical lattice*”. As in the previous reference, no real, e.g. organic, material context is invoked and the work focuses on cold-atom simulators, and here only as a theoretical proposal. The work rather focuses on pumping, which one cannot realise in the organic materials central

to our work, unlike in the simulators. In the protocol, only two specific pumping parameters correspond to realising the SSH model [which is cited as Ref. [31], original work by SSH, which we also included in our manuscript]. Once again, the physics here is non-interacting and addressed the quantum simulation in an optical lattice, and hence is not relevant to realising excitons from an electronic structure of a real material [hence cannot even simulate topological excitons, which we predicted].

Ref. [3] - M. Lohse, C. Schweizer, O. Zilberberg, M. Aidelsburger, and I. Bloch, A Thouless quantum pump with ultracold bosonic atoms in an optical superlattice. Nat. Phys. 12, 350 (2016).

This reference is about realising a “*Thouless quantum pump with ultracold bosonic atoms in an optical superlattice*”. Therefore, this work is about realising a Chern invariant in a two-dimensional parameter space, rather than focusing on the SSH physics of the original model that is pumped. Once again, this pumped simulator cannot simulate the interacting SSH physics and topological excitons central to our work. This is because the simulation suffers from the hardcore interaction “*Due to the large on-site interaction, each atom is localized on an individual double well*”, and there is no interaction at longer range, which can realise the necessary binding interactions U_1/U_2 of the interacting model central to our work. It should be stressed again, that the pumping/simulation context [rather than material/electronic context], and the lack of appropriate interactions in the experiment for even a simulation of the excitons, allows us to conclude that this reference is irrelevant to our work.

Ref. [4] - S. Nakajima et al., Topological Thouless pumping of ultracold fermions. Nat. Phys. 12, 296 (2016).

Here, we can be rather brief and explicit. This reference did not even cite any of the Su-Schrieffer-Heeger (SSH) original papers, and therefore deems SSH model as irrelevant to that published work. Instead, it focuses on realising the Thouless pump and the Chern invariant, without any reference to SSH physics. This shows that this reference is not only irrelevant to the SSH model considered in our work, but also proves the point that the Thouless pump (central to the other cold-atom simulation references brought up by the Referee) is an independent entity that can be realised without referring to SSH model. Hence, respectfully, we firmly stand by the opinion that there is no point referring to this paper by both us (authors) and the Referee, as we do not consider any kind of Thouless pump (nor the Chern invariant) in our work.

Ref. [5] - M. Leder et al., Real-space imaging of a topologically protected edge state with ultracold atoms in an amplitude-chirped optical lattice. Nat. Comm. 7, 13112 (2016).

In this reference, the authors “*experimentally realise an interface between two spatial regions of different topological order in an atomic physics system*”. They “*directly observe atoms confined in the edge state at the intersection by optical real-space imaging and characterize the state as well as the size of the associated energy gap*”. This work is irrelevant to our paper, as just like the other papers cited by the Referee, it focuses on the quantum simulation rather than real material context of the SSH model. Moreover, it also does not realise any of the excitonic, or in particular topological excitonic physics, whatsoever. Instead, the work focuses on simulating the domain wall problem, which only simulates *non-interacting* electronic (rather than excitonic) edge states from the localised free single fermionic (here, in experiment, rubidium atoms ^{87}Rb are also fermionic!) particles. In that regard, it should be stressed that unlike the atoms considered in any of the mentioned experiments, the topological exciton is a composite bosonic (and extended, as we show with our quantum-geometric bounds) object/quasiparticle, consisting

of a superposition of *interacting* electrons and holes.

Ref. [6] - Y.-G. Peng et al., Experimental demonstration of anomalous Floquet topological insulator for sound. Nat. Comm. 7, 13368 (2016).

This reference reports an “*experimental demonstration of anomalous Floquet topological insulator for sound*”. In this case, we were most confused. In our work, we not only do **not** consider any Floquet physics at all, but also focus on excitons and the physics of interacting electrons and holes, rather than sound. This work not only does not mention any of the SSH physics, neither cites any of the original SSH papers, neither alludes to any materials that could realise SSH physics. Instead, it focuses on simulating non-interacting anomalous Floquet topological insulators in acoustic metamaterials. Therefore, this work cannot be deemed relevant to any of the (undriven) organic semiconductor/material modelling context of our work, neither to any of the interacting SSH physics realised by our titular excitons (as no interacting Hubbard-like terms were simulated with sound here). Moreover, as we stressed, we do not consider any of the driven aspects in our work, as we consider topological excitonic bound states already generated and stabilised with the interactions, therefore the non-interacting Floquet physics is completely irrelevant.

Ref. [7] - A. Gómez-León and G. Platero, Floquet-Bloch theory and topology in periodically driven lattices. Phys. Rev. Lett. 110, 200403 (2013).

This reference is about Floquet theory and topology in periodically-driven system. The authors claim to provide “*a general framework to solve tight binding models in D dimensional lattices driven by ac electric fields*”. While the work is certainly interesting (and highly-cited), it concerns the physics of **only** non-interacting particles, therefore it cannot realise any excitons relevant to our work. The excitons are definitionally bound states of electrons and holes formed under attractive interaction, and **no** interactions are considered in the provided reference. Moreover, no connection to real materials is made in this Letter, proving that this work is irrelevant to our manuscript in that aspect as well. Finally, the reference alludes only to the “*ac driven*” dimer (SSH) model [citing only one of the original SSH papers – “Ref. [15]” – contrary to our work, where we credit both of the original works of SSH]. Therefore, this work alludes to the SSH model only in the free particle case *driven by the ac field*, whereas in our work, no drive is considered, but instead we focus on the interacting physics of the bound states of electrons and holes (i.e. excitons). That allows us to conclude that this work is not only irrelevant to the titular excitonic aspects of our work, but is also irrelevant to the topological physics of **undriven** electrons [modelled with the standard SSH model without ac-drive], before the interactions were even considered. Therefore, we are deeply confused why the Referee(s) would expect us to cite this work, given its irrelevance to both electronic and excitonic aspects of our manuscript [in strong contrast to the fact that, unlike this Reference, we actually credited the original (undriven) SSH works more appropriately]. It comes even more to our surprise, given we do not consider any of the Floquet physics in our work. Instead, we rather focus on topological excitons, which are possible only if the interactions of the fermionic degrees of freedom (electrons and holes) are present.

Ref. [8] - V. Dal Lago, M. Atala, and L. E. F. F. Torres, Floquet topological transitions in a driven one-dimensional topological insulator. Phys. Rev. A 92, 023624 (2015).

This manuscript considers “*Floquet topological transitions in a driven one-dimensional topological insulator*”. The reason of irrelevance to our work is similar to the case of the Reference above. Only

non-interacting physics under Floquet drives is considered, with the latter not being considered in our work whatsoever. At the same time, the former (non-interacting models considered) is unable to generate the titular topological excitonic physics, unlike our interacting models, which further shows the irrelevance of this work to our material.

Ref. [9] - F. A. An, E. J. Meier, and B. Gadway, Engineering a flux-dependent mobility edge in disordered zigzag chains. Phys. Rev. X 8, 031045 (2018).

In this reference, the authors “*report the first cold-atom observation of a one-dimensional non-interacting mobility edge*”. Once again, all the physics is non-interacting, and hence cannot realise excitons titular and central to our work. Moreover, rather than concerning real materials (as our work does), this work focuses on cold atom quantum simulators. Moreover, none of the original works by Su-Schrieffer-Heeger are cited in this paper, showing that this work is also not concerned with the SSH physics aspects of the electronic degrees of freedom introduced in our work – even before the interactions could be in-principle used to allow a formation of topological excitons, which is central to our manuscript.

Ref. [10] - B. Perez-Gonzalez, M. Bello, A. Gomez-Leon, and G. Platero, Interplay between long-range hopping and disorder in topological systems. Phys. Rev. B 99, 035146 (2019).

This reference is about including long-range hopping and disorder in the standard SSH model. In that context, the authors “*analyze how the electronic and topological properties are affected*”, with no relation to (or even mentioning of) the excitonic physics. Even more so, the generalized Hamiltonian considered in this work is manifestly non-interacting, therefore, unlike our models, cannot realise any excitonic physics by construction, showing further irrelevance to the problem of excitonic topology studied in our work. On an additional, although not equally relevant note, the authors claim that “*they discuss implications (of the results concerning the extended non-interacting model) in realistic transport measurements*”, but (unlike in our work concerning organic semiconductors) no particular real material is mentioned. Instead, only the relevance to simulators (e.g. trapped ions, cold atoms) is discussed, besides quantum dots of the Ref. [37] cited in the paper. Notably, in the Ref. [37] itself – no relation to SSH physics was claimed.

Consequently, the theoretical framework proposed in the study needs to be validated, which is essential to confirm the feasibility and practical relevance of the findings. To be concrete, why can the SSH model describe the single-particle electron and hole states of polyacene (Eq. 5) without needing to consider the geometry of polyacene?

We are again puzzled by this comment, as we extensively validated our model in the original manuscript and explicitly included the effects of the geometry of the polyacenes.

In detail, we performed first principles density functional theory calculations for a range of polyacenes, with our DFT results showing good agreement with previous studies of *electrons* in the same system [Phys. Rev. B 106, 155122, Nat. Nano. 15, 437-443]. In these calculations, we directly described the ring geometry of the acenes as it is directly related to the changes in the electronic structure that drive the transition from trivial to topological phases in both electrons and excitons. Using these results, we parametrized the SSH model, which provides a good description of the band edges of the polyacenes. To make this more explicit, we include a direct comparison of the SSH bands with the DFT bands in Fig. 1 below.

Figure 1. Bandstructure for DFT (red) and fit using SSH model (blue) for (a) polyanthracene and (b) polypentacene.

2. Controllability of Excitonic Topology: The controllability of excitonic topology through chemical functionalization and dielectric environment is highlighted in the abstract. A more in-depth analysis of the mechanisms and techniques for achieving this control in the main part would enhance the practical relevance of the research.

We thank the Referee for their comment, and we agree that experimental aspects are important. However, we are somewhat confused by this comment as the manuscript described in detail both chemical functionalisation and dielectric environment. Figure 2 of the main text shows the different topological exciton phases hosted by different chemical compositions ($N = 3, 5, 7$). In the revised manuscript, we show that the effects of dielectric screening induced by different substrates demonstrate a robustness of the excitonic topology. These results are then discussed in detail in the text. In the revised version, we have also extended our analysis with a further quantitative calculation of how the quantum geometry of excitons can be controlled with the dielectric environment, while satisfying the topological bound (Fig. 4 in Supplementary Information). As a second new addition in the revised manuscript, we have also included an additional analysis of the controllability of the excitonic topology with externally applied strain fields. Very importantly, these all provide for practical mechanisms to control the excitonic topology. We hope that these clarifications and updates fully resolve the Referee's request concerning the practical relevance of our research. We note that in previous works on the electronic topology in these materials, moving from the polymer in a gaseous phase (freestanding) to a metallic substrate led to a closing of the electronic band gap [Cirera et al Nat. Nano. 15,6 2020]. Such bandgap renormalisations are beyond the scope of this work, but further speak for the broader tunability of the electronic and hence excitonic topology of polyacene polymers.

3. Clarification on Figure 3: Is Figure 3 a schematic diagram only, or is it numerically computed? If it is numerically computed, what parameters are used in each subfigure?

Yes, Fig. 3 in the main text is a schematic illustration comparing the SSH physics of *electrons* (pictorial Wannierisation of the model originally introduced by Su, Schrieffer, and Heeger) and the shifts of the *excitonic* Wannier states due to the excitonic invariant. Only the ratio between hoppings and interaction terms are relevant for this illustration. We have revised the Figure to further expose its schematic nature. The schematics of the Figure coincide with the quantitative results, which are detailed in both main text and Methods.

4. Benefits of Introducing Quantum Geometry: What benefits do we get from introducing quantum geometry (the imaginary part of which recovers Berry curvature)? Figure 2a and Figure 4b seem to indicate that they carry the same information. Is the quantum geometric tensor more experimentally accessible through the spatial spread of the states, for example?

Quantum geometry is an increasingly useful tool in the evaluation of topological physics. The benefit in our case, as explained in the main text, is that we can use geometry to set a lower bound on the spread of the exciton wave function.

Generally, bounds applicable to response theories can be directly phrased in this geometric language. It is for this reason why research in van der Waals materials [*Nature* 614, p. 440–444 (2023)] and superfluids [*Nature Communications* 6, 8944 (2015)] and other interacting topological systems [*Nature Physics* 20, 1262 (2024)] is benefiting from this view and we show here that it is similarly powerful in the context of excitons. Using this geometric framework we show that non-trivial excitons exist and that their wave function features are set by bounds that are derived from it. We additionally show that these fundamental bounds are consistent with numerical DFT results in real organic materials, underpinning the exciting nature of our results. The quantum geometric tensor is indeed experimentally accessible in diffusive transport and optical (manifestly, non-adiabatic/interband) responses, with their matrix elements reflecting the spatial spreads of the states. Both mentioned types of responses are central to the engineering of excitons in real materials. Indeed, we pursued [arXiv:2410.00967], and further plan to pursue these aspects and questions as the subject of future studies, on utilizing the well-established connections between the quantum geometry and responses of quantum states defined in more general Hilbert spaces, independently of whether these concern electronic, magnetic, or excitonic degrees of freedom.

5. Lower and Upper Bounds of Excitonic Spread: Equation 10 gives a nontrivial lower bound for the topologically nontrivial state. Is there any upper bound for the topologically trivial state? If there is no bound, can a trivial state also have a large spread?

We thank the Referees for this question. There is no upper bound for the topologically trivial state: topological excitons can have an arbitrarily high spread, as long as it satisfies the quantum-geometric lower bound. And as suggested by the Referee, one could indeed encounter a situation with a topologically-trivial state that has a large spread. In fact, while we discussed this possibility previously in the Methods, we find it to not be the case in the organic materials studied. Intuitively, the presence of spread in the maximally-localised excitonic Wannier functions is equivalent to the enhancement of the metric, as we analytically derived. The presence of a comparably enhanced quantum metric without any non-trivial topological invariant bounding it from below, while not impossible, is expected to be rare, as the metric is explicitly

given by the elements of the non-Abelian Berry connection. The presence of enhanced non-Abelian Berry connection is known to reflect the non-triviality of Wilson loops [Phys. Rev. B **100**, 195135 (2019)]; yet, the non-triviality of the Wilson loop means that there is a topological invariant underneath the non-Abelian Berry connection and associated momentum-space quantum metric content.

An important consequence of excitonic spread concerns excitonic transport [arXiv:2410.00967], which can be experimentally measured. Here, the lower-bounded quantum-geometric spread of topological excitons stands in strong contrast with the sharply-localized trivial excitons. Even though the excitonic band structures of two types of excitons are identical, the quantum geometry underpinning distinct minimal spreads results in manifestly different diffusive and driven transport signatures.

6. From the materials point of view, the spin of the exciton is one of the most important characteristics defining the most important optical properties of organic semiconductors. In this connection, it is important to include the spin into consideration. Unfortunately, from the current study, it is not intuitive whether including the spin will preserve the band topology, particularly the part related to the Wannier envelope of the wave function. The authors mention the relevance of the excitonic spin in the opening paragraph of the Discussion section, but this important issue must be discussed more thoroughly.

We agree with the Referee that spin is an important ingredient in discussing excitons, and indeed we had already highlighted this fact in the Discussion section of our original manuscript as planned future work with a particular focus on optical aspects. We now further extend the Discussion of the spin aspect in the revised manuscript; yet, the role of spin for the excitonic topology and geometry in the considered organic materials is to be deemed irrelevant, as we explain in detail below.

Briefly, the (excitonic) topological classification described in our manuscript is independent of the spin state of the system, with both singlet and triplet excitons satisfying bosonic time-reversal symmetry, i.e. $\mathcal{T}^2 = +1$. More concretely, in systems with weak spin-orbit coupling (such as most organics), spin is a good quantum number and excitons appear as triplet or singlet excitons. Their Hamiltonians differ in one term only: singlet excitons experience an extra exchange interaction that is not present for triplet excitons [see, e.g. Phys. Rev. **B** 62, 4927 (2000)]. While this distinction will lead to quantitatively different envelope functions for the singlet and triplet states, the topology analysis presented in our work is equally applicable to either type of envelope function, as it relies on the qualitative/topological, rather than on particular quantitative, features. In other words, in the one-dimensional materials considered in this work, the interaction potentials for both cases of spin species lead to the same excitonic topology and envelope/single-particle contributions to the topological invariant. More particularly, we find that the envelope contribution in the one-dimensional polyacenes is always, and independently of spin, *trivial* under the presence of inversion and time-reversal symmetries (see Supplementary Information). This means that both singlet and triplet excitons realised in the material would host the same topological invariant defined under the necessary inversion symmetry. An interplay of the singlet and triplet excitons hosting the same excitonic topology in the considered materials, as well as its connection to optics and experimental controllability, deserve further study.

References

[1] M. Atala et al., Direct measurement of the Zak phase in topological Bloch bands. Nat. Phys. **9**, 795–800 (2013).

- [2] L. Wang, M. Troyer, and X. Dai, Topological charge pumping in a one-dimensional optical lattice. *Phys. Rev. Lett.* 111, 026802 (2013).
- [3] M. Lohse, C. Schweizer, O. Zilberberg, M. Aidelsburger, and I. Bloch, A Thouless quantum pump with ultracold bosonic atoms in an optical superlattice. *Nat. Phys.* 12, 350 (2016).
- [4] S. Nakajima et al., Topological Thouless pumping of ultracold fermions. *Nat. Phys.* 12, 296 (2016).
- [5] M. Leder et al., Real-space imaging of a topologically protected edge state with ultracold atoms in an amplitude-chirped optical lattice. *Nat. Comm.* 7, 13112 (2016).
- [6] Y.-G. Peng et al., Experimental demonstration of anomalous Floquet topological insulator for sound. *Nat. Comm.* 7, 13368 (2016).
- [7] A. Gómez-León and G. Platero, Floquet-Bloch theory and topology in periodically driven lattices. *Phys. Rev. Lett.* 110, 200403 (2013).
- [8] V. Dal Lago, M. Atala, and L. E. F. F. Torres, Floquet topological transitions in a driven one-dimensional topological insulator. *Phys. Rev. A* 92, 023624 (2015).
- [9] F. A. An, E. J. Meier, and B. Gadway, Engineering a flux-dependent mobility edge in disordered zigzag chains. *Phys. Rev. X* 8, 031045 (2018).
- [10] B. Perez-Gonzalez, M. Bello, A. Gomez-Leon, and G. Platero, Interplay between long-range hopping and disorder in topological systems. *Phys. Rev. B* 99, 035146 (2019).

For our detailed responses on all the above references brought to our attention by the Referee(s), please see our response to Referee's point "1".

Reviewer 2 (Remarks to the Author):

The authors perform DFT calculations on various polyenes and model their electronic properties with a tight-binding approach characterized by two hopping per unit cell t_1 and t_2 . Additionally, they obtain the electronic orbitals essential for determining the excitonic bands of the system by solving the Wannier equation. Once the excitonic band structure is obtained the manuscript explores the topology and geometric effects of excitonic bands.

I have some doubts and questions on the results that the authors should clarify before publication.

We thank the Referee for taking their time to review our manuscript. We welcome their questions as an opportunity to further clarify our material.

1) In Fig.2, the exciton energies are the same for the pair $\{t_1, t_2\}$ as they are for the pair $\{t_2, t_1\}$?

We thank the Referee for this important question. The exciton energies are indeed the same for both of the mentioned pairs. Such scenario is analogous to the *electronic* SSH model, where the band energies are identical across the entire momentum-space for different topological/trivial phases.

2) I don't understand why does not appear a phase IV, similar to phase III. In Fig.2 The exciton Hamiltonian possess spatial inversion symmetry. Due to this symmetry and the translational symmetry of the system, there are two inequivalent inversion centers: one at $x = 0$ and another at $x = a/2 \pmod{a}$. These correspond to the two possible locations of the exciton Wannier function center, as discussed by Zak (PRL 45, 1025). The Wannier center at $x = 0$ corresponds to an excitonic Zak (Berry) phase $\phi = 0$, while the center at $x = a/2$ corresponds to $\phi = \pi$ (Zak PRL 48, 359), both modulo 2π . In the studied system, the inversion centers are located at the centers of the bonds with hopping parameters t_1 and t_2 . If the origin of coordinates is shifted from the center of a bond with hopping t_1 to the center of a bond with hopping t_2 , the center of the excitonic Wannier function shifts by $a/2$. Consequently, the Zak phase changes by π .

Thus, if a system with hopping parameters t_1, t_2 has a Zak phase of 0 and a Wannier center at 0, interchanging the hopping parameters should lead to an energy degenerated Zak phase of π and a Wannier center at $a/2$.

Therefore, in this system, I would expect that the existence of a phase III should be accompanied by the existence of a complementary phase IV.

We thank the Referee for this important point, which has allowed us to identify a typo in one of our numerical equations (a sign typo). The resulting phase diagram is exactly consistent with the expectations of the Referee.

In more detail, prompted by the Referee's insights, we revised the numerical results and searched further for phase IV. We identified a sign error in the Wannier equation we used in our numerics, explicitly

$$W_{k,-k',Q} = V_{\text{NR}}(k - k') \sum_{i,j \in \{A,B\}} \varphi_{i,k+Q/2}^* \varphi_{j,k'-Q/2}^* \varphi_{j,k'+Q/2} \varphi_{i,k-Q/2},$$

has been corrected to:

$$W_{k,-k',Q} = V_{\text{NR}}(k-k') \sum_{i,j \in \{A,B\}} \varphi_{i,k+Q/2}^* \varphi_{j,k'-Q/2}^* \varphi_{j,k-Q/2} \varphi_{i,k'+Q/2}.$$

Having corrected the evident sign error, there is indeed a symmetry between phases III and phase IV, as the Referee suggests. In particular, both do *not* exist in the studied materials, and the revised phase diagram only includes the trivial regime I and the topological regime II. Just as in the case of the *electronic* SSH model, regime I and regime II realise distinct quantum geometries associated with the second moments of the maximally-localised Wannier functions, and therefore the rest of our analysis holds.

The fundamental distinction between the two inversion-symmetry phases in the Referee's argument is the atomic/molecular limit which one uses as a starting point. What this means for the general model in terms of t_1 and t_2 without reference to specific material realisation, is that on decoupling to an atomic/molecular limit ($t_2 \rightarrow 0$), the spread of the electronic wavefunctions can become arbitrarily small, and is not topologically-bounded. The topologically-bounded spread emerges as a consequence of a stronger intermolecular coupling, as compared to the intramolecular binding/localisation captured by the intramolecular hopping amplitude t_1 in the atomic/molecular limit. The intermolecular coupling t_2 introduces the quantum-geometric obstruction, analogously to the electronic geometric obstruction introduced by the hopping t_2 between the dimers in the electronic SSH Hamiltonian, cf. [*Phys. Rev. Lett.* **124**, 167002 (2020)], as an example.

We further clarify the absence of region IV (and of region III as well) in the updated text.

I suspect that, in Fig.2, when the hopping amplitudes are interchanged the form and position of the electron and hole wavefunction are not interchanged. And therefore this model are not exactly equivalent to the SSH model, where two degenerated topologically different phases appear.

We thank Referee for this point. As justified in the previous point, given the present atomic/molecular limit (associated with two hybridized molecular orbitals at the polyacene rings), there is a crucial geometric difference in the second-moments of electronic/hole functions, when changing t_1 and t_2 , which induces the non-trivial topological invariant, and the unorientability of the Bloch bundle. That distinction, then retrieves the connection to the interacting SSH model where not only the relative values of t_1, t_2 are important, but the interaction strengths U_1, U_2 can break the symmetry with their specific values within the model, when t_1 and t_2 are swapped. Importantly, swapping t_1 with t_2 , and *simultaneously*, U_1 with U_2 , retrieves two degenerated topologically different excitonic phases (maps phase I to phase II). On the other hand, to obtain phase III and phase IV, one would require to only swap the hoppings, or the interactions individually. With the corrected numerics, we ultimately show that such a counterintuitive, but in principle possible, inversion does not occur in the polyacenes under the ambient conditions.

If this is the case, which orbitals are used for each pair of t_1, t_2 ? DFT calculation only provide this information for a small number of polienes?

We thank the Referee for this important point. The used orbitals are constituted by the weighted linear combinations of p_z orbitals contributing to the molecular orbitals over the rings of polyacenes, as shown in Fig. 6. We further clarify this in the updated text (Methods). Above, we also clarified that the case is

indeed different, with the SSH-like energetic equivalence of phases I and II.

3) The Berry (Zak) phase is calculated using Eq. 14. The system has inversion symmetry. Is it possible to simplify this calculation by using the parity eigenvalues at the Time-Reversal Invariant Momenta (TRIMs), such as Γ and X ? (Fu-Kane PRB 76 045302 2007).

We thank Referee for this important question. Our approach follows (Hughes *et al.* PRB 83, 245132 2011), where the inversion invariant is computed from the Berry connection. The reason for our choice is that it provides a direct connection to quantum geometry. The Referee is correct that it is possible to compute the invariant from the inversion eigenvalues at TRIMs, and that such an approach is simpler for characterising the topology itself. Yet, this alternative approach has no direct connection to quantum geometry. Nevertheless, following the Referee's constructive suggestion, we have added a comment on this point in the updated manuscript. Namely, we clarify how the invariant can be evaluated from the parity eigenvalues, by writing:

Moreover, the invariant can be alternatively deduced from the high-symmetry points (HSPs) $Q = 0$ and $Q = \pi$, analogously to how the topological invariants can be deduced from time-reversal invariant momenta (TRIMs) in the electronic topological insulators [Fu & Kane (2007)]. Equipped with the parities of the excitonic band at the HSPs, δ_i , the excitonic invariant satisfies,

$$(-1)^{P_{exc}} = \prod_i \delta_i. \quad (1)$$

Hence, $P_{exc} = 1$ manifestly requires different band parity eigenvalues at $Q = 0$ and $Q = \pi$.

In the above, we further refer to the work by Fu and Kane [Phys. Rev. B **76**, 045302 (2007)], where the parity eigenvalues at the high symmetry points (HSPs), here also equivalent to TRIMs, were introduced to indicate the inversion-symmetry protected topology in the electronic contexts.

4) Did you calculate the values of U_1 and U_2 at different points in the phase diagram, or did you estimate the values of U_1 and U_2 based on the form of the phase diagram?

We thank the Referee for this question which allows us to clarify our methodology. The construction of the phase diagram in Fig. 2 of the main text is not based on the simple SSH-Hubbard-like model in which the U_i interaction terms feature. Instead, that phase diagram is based on the full Wannier equation solution, in which all the interaction terms U_i , of arbitrarily high range, are effectively included.

Our discussion of the SSH-Hubbard-like model with U_i interaction terms is only used as a phenomenological description of the full system to explore the limits of strong and weak interaction. This allows us to provide a qualitative description of the various regimes arising from the more accurate Wannier equation numerical calculations presented in Fig. 2. As such, the values of U_1 and U_2 were not explicitly calculated.

In the revised manuscript we have re-organised the discussion to clarify that we use the full Wannier equation for Fig. 2, and the SSH-Hubbard model only as a phenomenological description to qualitatively understand the different regimes obtained numerically.

5) How were the parameters t_1 and t_2 obtained from the DFT calculations? Did you use Wannier functions to derive these parameters, or did you simply fit the last two subbands using two hopping parameters?

Plotting the tight-binding band structure over the DFT results could help assess the quality of the approximation.

We thank the Referee for this question, which allows us to clarify our approach. In our original manuscript, we used a direct fit to the two band edges around the gap to determine t_1 and t_2 . We show a comparison of the SSH bands and the DFT bands in Fig. 2.

Figure 2. Bandstructure for DFT (red) and fit using SSH model (blue) for (a) polyanthracene and (b) polypentacene.

6) In Fig. 7, you plot the excitonic bands for different organic semiconductors. In cases (b) and (c), it appears that the bands are degenerate at the X points. If this is the case, the bands are not isolated, and it would not be possible to compute the Berry phase for individual bands. Could you discuss on this?

We thank the Referee for this point, which helps us improve the clarity of our manuscript. The bands are actually isolated, as in case (a), although this was not obvious in the original figure, as the Referee highlights. As can be seen in the updated Figure, which, importantly, was obtained on solving the *corrected* Wannier equation, the excitonic bands are well-isolated. As a result, there are no problems for the excitonic Berry phase evaluation.

7) The spread of the exciton Wannier function is a gauge-dependent quantity, as discussed in Vanderbilt's book. Which gauge did you use in your calculations? Did you try to minimize the size of the Wannier function?

We thank the Referee for this very important point. Indeed, the derivation of the connection between the spread of the exciton wave function and the excitonic quantum geometry requires the minimisation of the size of the associated Wannier functions. While, in general, the spread of the excitonic Wannier function is

a gauge-dependent quantity, the *minimised* spread of the maximally-localized excitonic Wannier functions (MLXWF) is a gauge-independent quantity. In turn, the quantum metric itself, which we connect to the spread of the MLXWF, is also a gauge-independent quantity. We have clarified this point in the revised manuscript/Supplementary Material.

8) I believe there is a typo, in the definition of $d(k)$ after Eq.20 in the Appendix. The sin and the cos seem to be interchanged.

We thank the Referee for spotting this important typo. Indeed, we accidentally interchanged cosine and sine, when typesetting. We are very grateful to the Referee for carefully reading our manuscript.

9) In some sections the manuscript is difficult to follow. It is not clear which approximation is done in the calculations. Also, it is not clear if the discussion on the dualized model is based in numerical calculation. The similitudes and differences with the SSH model are not clear. In order to follow the manuscript, it is necessary to frequently refer to the appendices. Therefore, it would be very helpful to include specific references to the relevant appendices in the main text whenever they are utilized.

We thank the Referee for this request to improve clarity. We have re-organised the main text to clearly identify what levels of theory are used at each stage, which we hope will improve the readability of our manuscript. But briefly, the discussion of the dualised model is in support of the numerical calculation based on a full Wannier equation solution and complementary to the real material DFT results. We have also revised the specific references to the relevant appendices/Methods sections.

The authors should revise the manuscript to clearly outline the approximations employed at each stage of their analysis. Additionally, they should discuss the connection with the SSH model.

We thank the Referee for this important point. We have revised the references to the approximations employed at different stages of our analysis, as also mentioned above. We have further elaborated on the relevant details concerning the connection to the SSH physics in the updated main text and Methods.

We would like to once again cordially thank the Referee for their careful reading of our manuscript. We found their points very useful for the improvements and revision of our manuscript that helped us clarify important points and helped in increasing the accessibility of our manuscript. In particular, we once again thank for the previous points concerning the symmetry between phase III and phase IV, which allowed to pinpoint the error in the Wannier equation, and as a result, to correct the results and associated Figures.

Reviewer 3 (Remarks to the Author):

We thank the Referee for taking their precious time to review our manuscript, especially given their Early Career Researcher stage. For the detailed responses to Referee's questions, please see responses to the other reports.

Reviewer 4 (Remarks to the Author):

The authors present a theoretical and numerical analysis of the topological properties of excitons in a family of one-dimensional systems based on the famous SSH model. This is a very timely topic which is being addressed only very recently by few groups and has the potential to push our understanding of excitons to higher grounds.

We thank the Referee(s) for very positive feedback on the topic of our work and we are very pleased to learn about their constructive evaluation and recognition of its significance.

Having said that, I think the manuscript has room for necessary improvement:

We thank the Referee for their constructive criticism. We welcome their questions and critical points as an opportunity to further improve our work.

1) Since, as the authors state: “The invariant P_{exc} precisely distinguishes two different topologies which are not transversable from one to another without closing a gap in the excitonic bands”, an important issue regarding the transition from topologically trivial to non-trivial is the closing and opening of the gap, it would be important to see a more detailed analysis of the exciton bands. Now this analysis comes down to a single figure in the Methods section where nothing can actually be seen. In fact, according to the colours, no gap closing seems to occur from a) to b). What can be made out of this plot?

We thank the Referee for their important remark, which is related to a point by Referee 2. Our original presentation of the exciton bands in the cases (b) and (c) used thick lines, making gaps not visible, unlike in (a). However, as explained in response to Referee 2, there was also a sign error in the Wannier equation, which previously introduced the asymmetry between phases I and II. On eliminating the error, we observe the bands in panels (b) and (c) to be well-isolated, as was the case in (a). Having resolved the numerical sign-error issue, the presence of the topology-protecting gaps in the revised Fig. 7 becomes clear. In other words, it can be explicitly made out of this plot that the excitonic bands are gapped, and hence, carry well-defined excitonic Berry phases ϕ_{exc} . Based on the updated phase diagram (Fig. 2); on having corrected the sign error, we here find that P_{exc} can be trivialised only if the *electronic* gaps were closed and reopened in the studied materials, i.e. electronic Berry phases were trivialised first, and then the trivial topology of electrons and holes was inherited by the excitons on solving the Wannier equation.

2) Region III in Fig. 2b,c, where the excitons become trivial on account of the interactions, takes over region II when interactions are quenched through a dielectric screening. I am afraid I cannot understand the arguments on page 4 that presumably explain this. Since the “trivialization” of the excitons originates in the interactions, Shouldn’t region III shrink when the interactions are screened (weakened)? Is the change from long-range to short-range interactions responsible for that? Does region II disappear altogether for a hard-core interaction model (presumably representing a very strong screening? Is the Hubbard model in the flat-band limit mimicking this case? In summary, I think the discussion should attempt to separate the effects of the range and the strength of the interaction in a clearer way and its connection to the very unclear discussion.

We thank the Referee for their important questions. Having corrected the sign error, region III indeed

shrinks. More precisely, it completely disappears. The reason for its disappearance is that hoppings and interactions are not uncorrelated, i.e. in the considered materials, $t_1 > t_2$ implies $U_1 > U_2$, and vice versa. Intuitively, increasing a hopping locally, such that $t_1 > t_2$, accumulates the electron charge density across a bond. The charge accumulation results in an increased Coulomb interaction underpinned by the Hartree term $U_1 > U_2$, while the exchange interaction term in organic polyacenes is not of comparable order of magnitude, as we also mentioned and clarified to Referee 1. Hence, given t_1, t_2, U_1, U_2 are not independent, only regions I and II persist.

Overall, as suggested by the Referee, we include a major revision of our discussion (and Methods) to reflect all of the raised very important points, which we elaborated on above (and now address in the updated manuscript, in particular focusing on the aspects of the range and strength of the interaction necessary for realising distinct excitonic topologies).

3) The final discussion contains too many promises, but no actual recipe for an experimental test of the topological properties described in the text. For instance, if an overall gap appears in the Brillouin zone one may expect localized excitons on the ends. However, I cannot imagine how this gap can appear judging from the bands presented. In my opinion, despite the very interesting theoretical insight gained, a likely or possible experimental manifestation of the predicted non-trivial topology would be necessary to make this work worth publishing in Nature Comms.

We thank the Referee for their important criticism, which we address in our revised manuscript. Explicitly, we revise the discussion to expand our ideas on the possible experimental manifestations. These involve details on the observation of the localized edge excitons in the topological regimes, which can be attained by local optical conductivity measurements (cf. updated Discussion). Furthermore, we further elucidate the promises on transport properties, which is the subject of a very recent follow-up preprint of ours that was stimulated by this work [arXiv:2410.00967]. In this recent preprint, we explicitly confirm our expectation of enhanced transport in topological excitons compared to their trivial counterparts, which we find holds in all transport regimes (free, phonon-limited, and polaronic). We also show that non-uniform electric fields can be used to directly probe the quantum metric of excitons, providing an experimental window into a basic geometric feature of quantum states. More technical details can be found in our mentioned, more recent, extended study of quantum-geometric manifestations in the transport of topological excitons mentioned above [arXiv:2410.00967].

We view the present work as setting the basics of exciton topology and quantum geometry. Our plan is to explore the various research directions suggested by this work and included in the Discussion over the next few years, and indeed our recent transport preprint [arXiv:2410.00967] is the first step in that direction. Similarly, we hope that our work will motivate others (theorists and experimentalists) to further explore exciton topology in organic materials and beyond.

We once again thank the Referee(s) for their positive feedback on the topic and significance of our work, and for the very constructive criticism, which allowed us to improve and revise our work.

Reviewer 5 (Remarks to the Author):

We thank the Referee for taking their precious time to review our manuscript, especially given their Early Career Researcher stage. For the detailed responses to Referee's questions, please see responses to the other reports.

Reviewer 1 (Remarks to the Author):

The authors have improved the clarity of their manuscript. Specifically, they have clarified the role of the SSH model and expanded their discussion of the experimental relevance of strain-engineered excitons in the supplementary materials, etc.

We thank the Referee for taking their time to review our manuscript. We further thank them for their affirmative comment on the improvement of the clarity of our manuscript. We welcome the Referee's further comments as an opportunity to further increase the accessibility of our material.

However, a few concerns persist:

Below we address the remaining comments of the Referee in a point-by-point manner, resolving the few persisting concerns with further clarifications.

1. While the authors have clarified how the ring geometry is incorporated into the DFT calculation and demonstrated the fitting of the SSH model parameters to the DFT results, this raises another critical question: Is fitting energy alone sufficient to capture the topological nature of the state? Topological information is generally encoded not only in eigenenergies but also in wave functions, which demands further scrutiny.

We thank the Referee for their question. The fitting of energy alone is not sufficient to capture the topological nature of the state. The form of the wavefunction (as provided in Fig. 6, Methods) is necessary to show the topological nature of the state.

To resolve Referee's scrutiny, we note that topological wavefunctions of electrons in polyacenes (incl. topological polypentacene) were previously (and consistently with our results in Methods) retrieved in the experimental studies [Nature Nanotechnology **15**, 437–443 (2020)], see the corresponding Supplemental Material. This study, which also includes *experimental* demonstrations of the SSH-like topological nature of *electrons*, leaves it with no doubt that the energy fitting for polypentacene, combined with the corresponding topological DFT wavefunction parity inspection [cf. caption of Fig. 6] as consistent with the experimental work mentioned, allows to conclude that one is in a topological electronic regime, as it concerns the polypentacene case.

We finally stress that further to the conclusion that polypentacene is in a topological electronic phase (as confirmed also experimentally in Ref. [Nature Nanotechnology **15**, 437–443 (2020)]), as we also reflected in the phenomenological SSH-Hubbard model, the central and titular part of our work is to address the topology of *excitons*. The topology of excitons, as follows from the numerical solution to the Wannier equation and from the calculation of the excitonic Berry phases detailed in Methods, does not assume any additional energy fittings, once the *electron* topology of the electronic DFT wavefunctions was established. In this work, we show that the topology of excitons is intuitively *inherited* from the topology of electrons [which we also retrieved in Fig. 6] under Coulomb interactions, while the corresponding topology of electrons was already experimentally confirmed [Nature Nanotechnology **15**, 437–443 (2020)].

2. The authors have addressed the experimental aspects.

We thank the Referee for approving our response on the experimental aspects.

3. No further comments.

We thank the Referee for providing no further comments on the previously addressed point.

4. and 5 are interconnected: Equation 10 establishes that the spread of topological excitons is lower bounded by $a^2/4 P_{exc}^2$, while Equation 3 defines P_{exc} as an integral of the Berry connection over the Brillouin zone. This implies that the quantum geometry tensor primarily acts as an intermediate quantity in the derivation, and its relevance to the paper's core findings remains unclear. Additionally, the experimental implications for diffusive transport and optical responses are undermined by the fact that even a trivial state could exhibit a large spread. Although the authors argue that the non-Abelian Berry connection determines the metric, so the large spread of trivial states is rare, without a more quantitative description, this assertion is weak, more or less.

We thank the Referee for an additional comment on their points 4 and 5. We would like to clarify that it is precisely the quantum geometric tensor that defines the spread of topological excitons (with its real part), not the excitonic Berry phase (i.e., integral of the Berry connection). Hence, the quantum geometric tensor is a direct (rather than an intermediate) quantity; its integral is equal to the excitonic spread (ξ^2), which is the central result behind Eq. (10) mentioned by the Referee.

We further clarify our assertion about the large spread of trivial states being rare, which we now also include in the Supplemental Material. In this context, what is meant by *non-trivial* includes all of the topological (and obstructed) phases - as all of these phases satisfy quantum geometric conditions due to topological (incl. topological crystalline) invariants. Hence, the remaining aspect is to explicitly show why in the absence of all such topological invariants in a considered phase [which corresponds to the *trivial* case of interest], the minimal spread of trivial states [equivalent to their quantum metric that is unconstrained by any lower bound] is reduced, with any possible enhancements of the metric being unlikely.

This more formally, yet intuitively, follows from the definition of the quantum metric:

$$g_{xx}^{\text{exc}}(Q) \equiv \langle \partial_Q u_Q^{\text{exc}} | (1 - |u_Q^{\text{exc}}\rangle\langle u_Q^{\text{exc}}|) | \partial_Q u_Q^{\text{exc}} \rangle \equiv \langle \partial_Q u_Q^{\text{exc}} | \hat{Q} | \partial_Q u_Q^{\text{exc}} \rangle = \langle \partial_Q u_Q^{\text{exc}} | \hat{Q}^2 | \partial_Q u_Q^{\text{exc}} \rangle = \|\hat{Q} | \partial_Q u_Q^{\text{exc}} \rangle\|^2, \quad (1)$$

where we defined a projector $\hat{Q} = 1 - |u_Q^{\text{exc}}\rangle\langle u_Q^{\text{exc}}|$ and used its idempotency, $\hat{Q} = \hat{Q}^2$, and Hermiticity $\hat{Q} = \hat{Q}^\dagger$ relations on connecting to the final of norm. In the case of the trivial topologies, the Bloch states $|u_Q^{\text{exc}}\rangle$ do not wind (nor half-wind), hence $|\partial_Q u_Q^{\text{exc}}\rangle \rightarrow \mathbf{0}$, with $\mathbf{0}$ the norm-zero vector. Hence, $g_{xx}^{\text{exc}}(Q) = \|\hat{Q} | \partial_Q u_Q^{\text{exc}} \rangle\|^2 \rightarrow \|\hat{Q}\mathbf{0}\|^2 \rightarrow 0$, as the Frobenius norm of \hat{Q} is smaller than for the identity matrix 1 that sums over the projectors onto all bands, rather than onto a smaller number of bands, as $\hat{Q} = 1 - |u_Q^{\text{exc}}\rangle\langle u_Q^{\text{exc}}|$ does on subtracting $|u_Q^{\text{exc}}\rangle\langle u_Q^{\text{exc}}|$, i.e., \hat{Q} excludes the projection onto $|u_Q^{\text{exc}}\rangle$. This concludes the more formal justification of the assertion that in the trivial states, the minimal spreads (ξ^2) are rare, given that $g_{xx}^{\text{exc}}(Q) \rightarrow 0$ and $\xi^2 = \langle g_{xx}^{\text{exc}}(Q) \rangle$, with an average which reads $\langle \dots \rangle = \frac{a}{2\pi} \int_{\text{BZ}} dQ (\dots)$,

as employed in the main text [Eq. (10)].

6. No further comments.

We thank the Referee for providing no further comments on the previously addressed point.

7 An extra comment: The deferral of the full SSH-Hubbard Hamiltonian, which is used to calculate many results in the main text, to the supplementary materials undermines the manuscript's accessibility.

We thank the Referee for the extra comment. While we indeed decided to defer the full SSH-Hubbard Hamiltonian to the Supplementary Material, we note that we already included the key phenomenological argument in the main text. As in our previous response, we would like to stress that our findings follow directly from the first-principles (DFT + Wannier Equation) calculations, and given that, it is not a factual statement to stress that SSH-Hubbard Hamiltonian “is used to calculate many results in the main text”.

Nevertheless, following Referee's suggestion – rather than deferring the SSH-Hubbard Hamiltonian to the Supplementary Material – we now include it more directly in the Methods, which accompanies the full body of the main text. The presence of the SSH-Hubbard Hamiltonian in the Methods allows the reader to directly access the relevant details of the phenomenological model, although the most important details related to the calculations of our results concern the central first-principles calculations and were already included in the Methods sections. While the importance of the corresponding Section is only phenomenological, this should further increase the accessibility of our material.

Reviewer 2 (Remarks to the Author):

I am satisfied with the changes made in the revised manuscript. I'm happy to recommend that the manuscript be published.

We thank the Referee for taking their time to review our manuscript. We are grateful to the Referee for recommending the publication of our material.

Reviewer 3 (Remarks to the Author):

We thank the Referee for taking their precious time to review our manuscript, especially given their Early Career Researcher stage. For the detailed responses to Referee's questions, please see responses to the other reports.

Reviewer 4 (Remarks to the Author):

The authors have done a fair job answering all the questions and comments of the referees and the manuscript has been considerably improved. The error spotted in the review process was a serious one, which considerably affected the results, but it has been amended and everything makes more sense now. I recommend publication.

We thank the Referee for their time to review our manuscript. We are further grateful for Referee's positive feedback and approving our improvements. We cordially thank the Referee(s) for recommending the publication of our material.

Reviewer 5 (Remarks to the Author):

We thank the Referee for taking their precious time to review our manuscript, especially given their Early Career Researcher stage.

Reviewer 1 (Remarks to the Author):

The authors have thoroughly addressed the questions and comments of the report, and the paper now deserves publication.

We thank the Referee for taking their time to review our manuscript. We further cordially thank the Referee for their questions and comments, which were thoroughly addressed in our response, helping us to improve our manuscript. Moreover, we thank the Referee for clearly recommending our paper for publication and we are pleased that we were able to address all of their comments and questions.

Reviewer 3 (Remarks to the Author):

We thank the Referee for taking their precious time to review our manuscript, especially given their Early Career Researcher stage. For the detailed responses to Referee's questions, please see responses to the other reports.